# Dickkopf-3 links HSF1 and YAP/TAZ signalling to control aggressive behaviours in cancer-associated fibroblasts

Nicola Ferrari [1,8], Romana Ranftl[1], Ievgeniia Chicherova[1], Neil D. Slaven[2], Emad Moeendarbary[3,4], Aaron J. Farrugia[1], Maxine Lam[1], Maria Semiannikova[1], Marie C. W. Westergaard[5], Julia Tchou[6], Luca Magnani [2] & Fernando Calvo[1,7]

Aggressive behaviours of solid tumours are highly influenced by the tumour microenvironment. Multiple signalling pathways can affect the normal function of stromal fibroblasts in tumours, but how these events are coordinated to generate tumour-promoting cancer-associated fibroblasts (CAFs) is not well understood. Here we show that stromal expression of Dickkopf-3 (DKK3) is associated with aggressive breast, colorectal and ovarian cancers. We demonstrate that DKK3 is a HSF1 effector that modulates the pro-tumorigenic behaviour of CAFs in vitro and in vivo. DKK3 orchestrates a concomitant activation of β-catenin and YAP/TAZ. Whereas β-catenin is dispensable for CAF-mediated ECM remodelling, cancer cell growth and invasion, DKK3-driven YAP/TAZ activation is required to induce tumour-promoting phenotypes. Mechanistically, DKK3 in CAFs acts via canonical Wnt signalling by interfering with the negative regulator Kremen and increasing cell-surface levels of LRP6. This work reveals an unpredicted link between HSF1, Wnt signalling and YAP/TAZ relevant for the generation of tumour-promoting CAFs.

[1] Tumour Microenvironment Team, Division of Cancer Biology, The Institute of Cancer Research, London SW3 6JB, UK. [2] Department of Surgery and Cancer, Imperial College London, London W12 0NN, UK. [3] Department of Mechanical Engineering, University College London, London WC1E 7JE, UK. [4] Department of Biological Engineering, Massachusetts Institute of Technology, Cambridge, Massachusetts 02139, USA. [5] Center for Cancer Immune Therapy, Department of Hematology, Herlev Hospital, University of Copenhagen, 2730 Herlev, Denmark. [6] Abramson Cancer Center and the Department of Surgery, Perelman School of Medicine, University of Pennsylvania, Philadelphia, PA 19104, USA. [7] Instituto de Biomedicina y Biotecnologia de Cantabria, c/ Albert Einstein 22, E39011 Santander, Spain. [8] Present address: Astex Pharmaceuticals, 436 Cambridge Science Park, Cambridge CB4 0QA, United Kingdom. Correspondence and requests for materials should be addressed to F.C. (email: calvof@unican.es)

Stepwise acquisition of genetic alterations is crucial for the initiation of primary epithelial tumours. Yet, increasing evidence supports the notion that concomitant stromal changes play a critical role in cancer progression in many types of neoplasias[1,2]. Fibroblasts constitute a significant proportion of the stromal compartment in many solid tumours. As opposed to normal fibroblasts, which are generally anti-tumorigenic[3], cancer-associated fibroblasts (CAFs) present a pathological activated phenotype that enables them to influence tumour progression, dissemination and response to therapy through remodelling of the extracellular matrix (ECM) and signalling to cancer, endothelial and immune cells[4,5]. In CAFs, signalling pathways such as Heat-Shock Factor 1 (HSF1) and YAP/TAZ are activated in response to cellular stress and mechanical cues, respectively[6,7]. In turn, HSF1 affects signalling to cancer cells promoting tumour growth whereas YAP promotes cancer cell invasion and angiogenesis through remodelling of the ECM. Thus, each pathway is regulated by different mechanisms and controls a defined set of functions; whether these molecular events are interconnected to regulate the emergence of a fully activated CAF phenotype is not known.

Dickkopf (DKK) proteins comprise a conserved family of secreted negative regulators of β-catenin[8]. DKK1, DKK2 and DKK4 have been shown to antagonise Wnt-mediated β-catenin stabilisation by binding and down-modulating Wnt co-receptor LRP5/6[9,10]. In contrast, DKK3 does not interact with LRP5/6 and thus is not considered a true Wnt signalling antagonist. The role of DKK proteins in cancer is thought to be mainly tumour suppressive, as they are commonly downregulated in cancer cells and can negatively affect proliferation and survival. Yet, the role of DKK proteins in cancer stroma is still understudied.

In this study we uncover an unprecedented role for DKK3 in linking HSF1 and YAP/TAZ signalling. We demonstrate that DKK3 is a HSF1 target gene that promotes aggressive behaviours in CAFs by potentiating YAP/TAZ activity via canonical Wnt signalling. Mechanistically, we show that DKK3 promotes LRP5/6 activity by interfering with the negative Wnt regulator Kremen1/2.

## Results

**DKK3 expression levels in the tumour microenvironment**. The initial evidence suggesting that DKK3 may play a role in the regulation of the tumour microenvironment came from analyses of stromal gene expression in normal and cancerous tissues (Supplementary Figure 1a). DKK3 gene and protein expression is significantly upregulated in the tumour stroma in several types of cancers including breast, colon and ovarian (Supplementary Figure 1a and Fig. 1a–c). Stromal expression of other DKK genes across various cancer types was less consistent (Supplementary Figure 1a), suggesting that DKK3 is the only DKK factor commonly associated with cancer stroma. DKK3 protein levels in breast cancer (BC) stroma were significantly increased upon progression to more aggressive cancers, particularly in ER-negative BC (Fig. 1d and Supplementary Figure 1b). Furthermore, in ER-negative BC there was a significant association between high stromal DKK3 gene expression and poor outcome (Fig. 1e). Analysis of human BC tissues showed that stromal DKK3 expression was restricted to vimentin-positive cells (Fig. 1f), an expression pattern characteristic of CAFs[11,12]. To investigate the cell of origin of DKK3 expression, we isolated different cell populations of murine MMTV-PyMT mammary carcinomas (Supplementary Figure 1c&d, see Methods for details). Dkk3 expression was restricted to CAFs (Fig. 1g). Similar findings were obtained in a human setting[13] (Supplementary Figure 1e). Stromal DKK3 expression in human tumours positively correlated

with the expression of CAF markers ACTA2, FAP and COL1A2 (Fig. 1h), supporting a link between DKK3 and CAFs. As DKK3 was restricted to CAFs in tumours, we investigated the prognostic potential of whole tumour DKK3 expression. These analyses indicated a significant correlation between DKK3 gene expression and poor outcome in ER-negative BC patients, as well as in colon and ovarian cancer patients (Fig. 1i).

**DKK3 is a HSF1 target gene upregulated in CAFs**. To investigate DKK3 in tractable models, we isolated CAFs from murine mammary carcinomas and human breast, colon and ovarian cancers, as well as normal tissue counterparts (NFs). We observed consistent upregulation of DKK3 mRNA and protein levels in CAFs compared to NFs and cancer cells (Fig. 2a, b and Supplementary Figure 2a&b). Comprehensive analysis of these models indicated that DKK3 expression was stronger in FAP-positive CAFs (Supplementary Figure 2b-d) but not necessarily associated with DKK3 secretion (Fig. 2b). Noteworthy, the DKK gene DKK2 also showed a significant enrichment in the cancerous stroma of some tumour types (Supplementary Figure 1a). However, DKK2 expression was not restricted to CAFs in murine or human tumours (Fig. 1g and Supplementary Figure 1d&e), nor upregulated in murine or human CAFs when compared to NFs (Fig. 2a, b and Supplementary Figure 2b). In addition, no association between DKK2 expression and poor outcome was observed in ER-negative BC, colon and ovarian cancer patients (Fig. 1i). For these reasons, we decided to focus on studying the role of DKK3 in CAFs.

To investigate the mechanism leading to DKK3 upregulation in CAFs, we first analysed the ChEA Transcription Factor Binding Site dataset in the Enrich database[14,15]. This informed of 58 transcription factors that potentially bind the DKK3 promoter (Fig. 2c). Of these, HSF1 was the only factor to positively modulate DKK3 expression in perturbation assays[16]. This was further confirmed in two independent datasets involving Hsf1 deletion in fibroblasts[6,17] (Supplementary Figure 2e). HSF1 is a transcription factor that can be activated by stresses such as oxidative stress, nutrient-deprivation and protein misfolding[18], which are commonly found in the tumour microenvironment. HSF1 has been associated to aggressive CAF behaviours by its ability to drive a transcriptional programme that supports the malignant potential of cancer in a non-cell-autonomous way[6]. Silencing Hsf1 expression reduced Dkk3 mRNA and protein expression in CAFs (Fig. 2f, g). Furthermore, HSF1 activity strongly correlated with DKK3 gene expression in the stromal compartment of human tumours (Fig. 2h). Next, we performed chromatin immunoprecipitation using anti-HSF1 antibodies and extracts from murine NF and CAFs. Fig. 2i shows that Hsf1 was significantly more associated with the bona fide HSF1 target gene Rilpl in CAFs. Further, Hsf1 interacted with promoter and enhancer regions of the Dkk3 locus, and this association was significantly increased in CAFs.

**DKK3 modulates pro-tumorigenic functions in CAFs**. To investigate the functional relevance of DKK3 we silenced Dkk3 expression in murine CAF1 with 2 independent RNAi (Fig. 3a). Knocking-down Dkk3 did not affect the expression of genes associated to CAF phenotypes (Fig. 3b). However, the ability of CAFs to contract collagen gels, a measure of their matrix remodelling capacity, was dependent on Dkk3 expression (Fig. 3c). Decreased gel contraction in Dkk3-siRNA CAFs was associated with thinner collagen fibres (Fig. 3d) and a significant decrease in matrix stiffness (Fig. 3e). Similar findings were obtained in CAF5 (Supplementary Figure 3a), a CAF line that did not secrete Dkk3 (Fig. 2b). For long-term functional analyses, we generated Dkk3

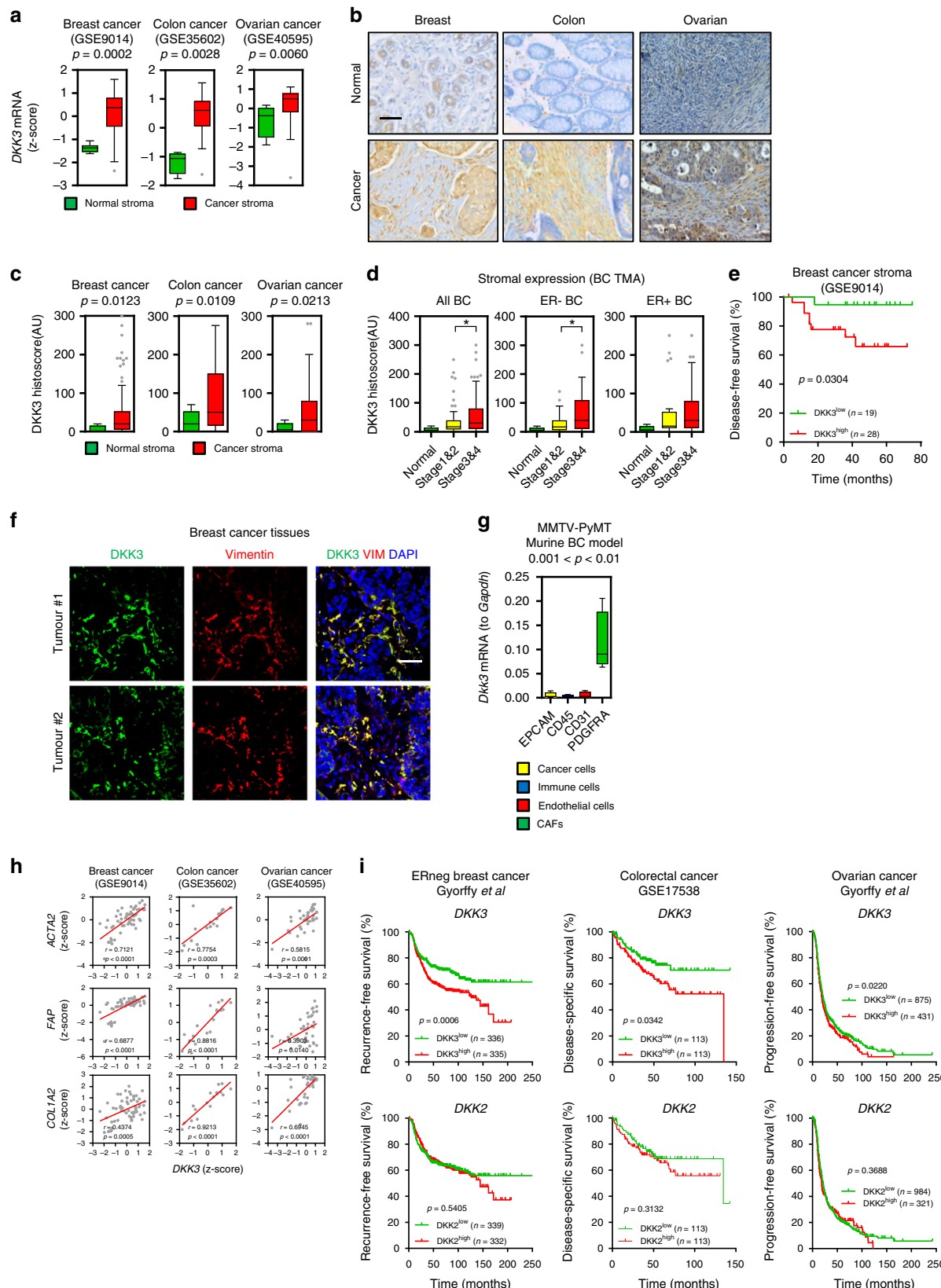

knockout CAFs (CAF-KO) and CAF-KO re-expressing Dkk3 (CAF-KO-REC)(Supplementary Figure 3b and Fig. 3f). Validating our previous results, Dkk3-null CAFs presented significantly less ECM remodelling activity (Supplementary Figure 3c, d). In three-dimensional (3D) co-culture models[19] (Fig. 3g), CAFs promoted cancer cell invasion and growth of murine MMTV-PyMT TS1 cancer cells, and this ability was dependent on *Dkk3* expression (Fig. 3h, i). These results were further validated using alternative approaches (i.e. matrigel on top) (Supplementary Figure 3e&f) and in human patient-derived CAFs from breast,

**Fig. 1** DKK3 is upregulated in the stroma of breast, colon and ovarian cancers. **a** Tukey boxplots showing z-score values of *DKK3* mRNA expression in normal and cancerous stroma from breast, colorectal and ovarian cancers (Breast: normal, $n = 6$; cancer, $n = 53$. Colon: normal, $n = 4$; cancer, $n = 13$. Ovary: normal, $n = 8$; cancer, $n = 31$). **b** Representative images of DKK3 staining in breast, colorectal and ovarian cancers and normal tissues. Scale bar, 100 μm. **c** Tukey boxplots showing quantification of DKK3 staining (Histoscore) in breast, colorectal and ovarian cancers and normal tissue counterparts (Breast: normal/adjacent, $n = 9$; cancer, $n = 109$. Colon: normal/adjacent, $n = 14$; cancer, $n = 107$. Ovary: normal/adjacent, $n = 8$; cancer, $n = 138$). **d** Tukey boxplots showing DKK3 Histoscore in non-invasive breast cancers (Stage 1&2), invasive breast-cancers (Stage 3&4) and normal tissue counterparts. Left graph shows all cancers irrespective of their subtype (normal, $n = 9$; Stage 1&2, $n = 74$; Stage 3&4, $n = 74$). Middle graph shows ER-negative breast cancers (normal, $n = 9$; Stage 1&2, $n = 40$; Stage 3&4, $n = 40$). Right graphs shows ER-positive breast cancers (normal, $n = 9$; Stage 1&2, $n = 21$; Stage 3&4, $n = 21$). **e** Disease-free survival of breast cancer patients stratified on stromal *DKK3* gene expression (GSE9014, ER-negative patients). **f** Images show DKK3 (green), vimentin (VIM; red) and DAPI (blue) staining of two representative human breast cancer tissues. Scale bar, 50 μm. **g** Tukey boxplot shows *Dkk3* mRNA expression levels (relative to *Gapdh*) in Cancer cells (Epcam[+]), immune cells (Cd45[+]), endothelial cells (Cd31[+]) and fibroblasts (Pdgfra[+]) from MMTV-PyMT tumours ($n = 4$). **h** Graphs show correlations between the expression of *DKK3* and *ACTA2*, *FAP* and *COL1A2* in normal and cancerous stroma from mammary gland (GSE9014), colorectal (GSE35602) and ovarian (GSE40595) human tissues. Pearson correlation coefficient (r) is shown. Each dot represents z-score values from individual patients. **i** Kaplan–Meier curves of recurrence-free survival, disease-specific survival and progression-free survival of ER-negative breast cancer, colorectal cancer and ovarian cancer patients, respectively, based on DKK3 and DKK2 gene expression. Where indicated, individual *p* values are shown; alternatively the following symbols were used to describe statistical significance: *$P < 0.05$; **$P < 0.01$; ***$P < 0.001$; #$P < 0.0001$; n.s., non-significant

colorectal and ovarian cancers (Supplementary Figure 4). Conversely, forcing DKK3 expression in NFs (Supplementary Figure 5a) significantly increased their abilities to remodel collagen matrices and promote TS1 invasion (Supplementary Figure 5b&c). In CAFs, HSF1 promotes tumour growth by inducing the production of secreted factors such as TGFβ and SDF1[6]. Using recombinant DKK3 and CAF-derived conditioned media from WT, KO and KO-REC CAFs we observed that secreted DKK3 had no effect on cancer cell proliferation and motility (Supplementary Figure 5d). Validating these results, we observed no significant differences in cancer cell proliferation or motility between cells treated with conditioned media from CAF1 (detectable secreted DKK3, as per Fig. 2b) or CAF5 (no detectable secreted DKK3), or in the presence of a blocking antibody against DKK3 (Supplementary Figure 5e). In addition, secreted DKK3 was not able to recover the functional defects associated to loss of DKK3 in CAFs (Supplementary Figure 5f). Thus, DKK3 is a novel HSF1 target gene that regulates ECM remodelling and associated cancer cell growth and invasion; unlike other HSF1 target genes such as TGFβ or SDF1, DKK3 does not have any apparent secreted function.

**DKK3 is required for the tumour-promoting activities of CAFs in vivo.** In vivo (Fig. 4a), we observed that tumours arising from cancer cells co-injected with KO-CAFs grew significantly more slowly than those of mice co-injected with WT-CAFs or KO-REC-CAFs, leading to increased survival (Fig. 4b, c). Conversely, ectopic expression of DKK3 in NF significantly increased their ability to promote tumour growth in vivo (Supplementary Figure 5g). Histological analysis revealed that TS1 + WT-CAF or TS1 + KO-REC-CAF tumours shared a poorly differentiated morphology and invasive phenotypes typical of high-grade tumours, with loss of basal lamina (i.e. laminin staining), despite having similar number of αSMA-positive CAFs (Fig. 5d). In contrast, TS1 + KO-CAFs tumours had a more differentiated architecture where cancer cell acini where surrounded by laminin-rich areas. Fibronectin and Masson's trichrome staining revealed increased ECM deposition at the tumour border in TS1 + WT-CAF and TS1 + KO-REC-CAF tumours, suggesting a higher reactive stroma. Second-harmonic generation (SHG) imaging confirmed these findings as TS1 + WT-CAFs tumours presented a higher content of collagen fibres than TS1 + KO-CAF tumours (Fig. 5f). These ECM characteristics have been linked with aggressive CAF phenotypes as they promote cancer cell motility and local invasion[20,21]. In agreement, we observed areas at the tumour border where cancer cells were invading in TS1 +

WT-CAFs tumours (Fig. 5f, asterisk). Using intravital imaging we confirmed this as co-injection of TS1 cells and WT-CAFs produced tumours where both cancer cells and CAFs were highly motile (Supplementary Movie 1; Fig. 5g, h). In contrast, motility in tumours with KO-CAFs was severely impaired (Supplementary Movie 2; Fig. 5h).

**DKK3 in CAFs potentiates Wnt/β-catenin and YAP/TAZ signalling.** To understand the molecular mechanism whereby DKK3 exerts its functions and how HSF1 connects to other signalling pathways, we performed global transcriptomic profiling of murine CAFs after *Dkk3* silencing. Gene set enrichment analysis[22] revealed a significant decrease in the expression of genes associated with Wnt/β-catenin and YAP/TAZ activities (Fig. 5a, Supplementary Figure 6a and Supplementary Table 1). β-catenin and YAP/TAZ are transcriptional regulators with paramount roles in development and cancer[23,24]. In CAFs, YAP establishes a transcriptional programme that enhances matrix remodelling and invasion of neighbouring cancer cells[7]. β-catenin plays a pivotal role in fibrotic disorders[25], but its relevance in modulating CAF functions is still not well understood. *Dkk3* knock-down decreased levels of non-phosphorylated (active) β-catenin and TAZ levels (Fig. 5b), reduced nuclear YAP/TAZ localisation (Fig. 5c) and inhibited β-catenin and YAP/TAZ transcriptional activity (Fig. 5d). These findings were confirmed using *Dkk3* knockout/recovery system (Supplementary Figure 6b&c) and patient-derived CAFs (Supplementary Figure 6d-f). Conversely, ectopic expression of DKK3 in human NFs increased non-phosphorylated (active) β-catenin and YAP/TAZ levels (Supplementary Figure 6g). Furthermore, we observed reduced β-catenin and YAP staining in DKK3-null CAFs in vivo (Fig. 5e, f), and DKK3 expression significantly correlated with YAP/TAZ and β-catenin signatures in human cancer stroma (Fig. 5g). We confirmed that β-catenin and YAP/TAZ were activated in murine and human CAFs (Fig. 5h and Supplementary Figure 7a&b), and in the stroma of breast, colorectal and ovarian cancers (Supplementary Figure 7c).

**DKK3 regulates CAF functions via YAP/TAZ.** Using our battery of functional assays we tested whether DKK3 was regulating CAF functions via YAP/TAZ and/or β-catenin. Contrary to YAP/TAZ silencing, knocking-down β-catenin (Fig. 6a) had no effect in the ability of CAFs to contract collagen-rich matrices (Fig. 6b), or promote cancer cell invasion and growth in 3D (Fig. 6c, d). Loss of YAP/TAZ function in CAFs leads to MLC2 and Src

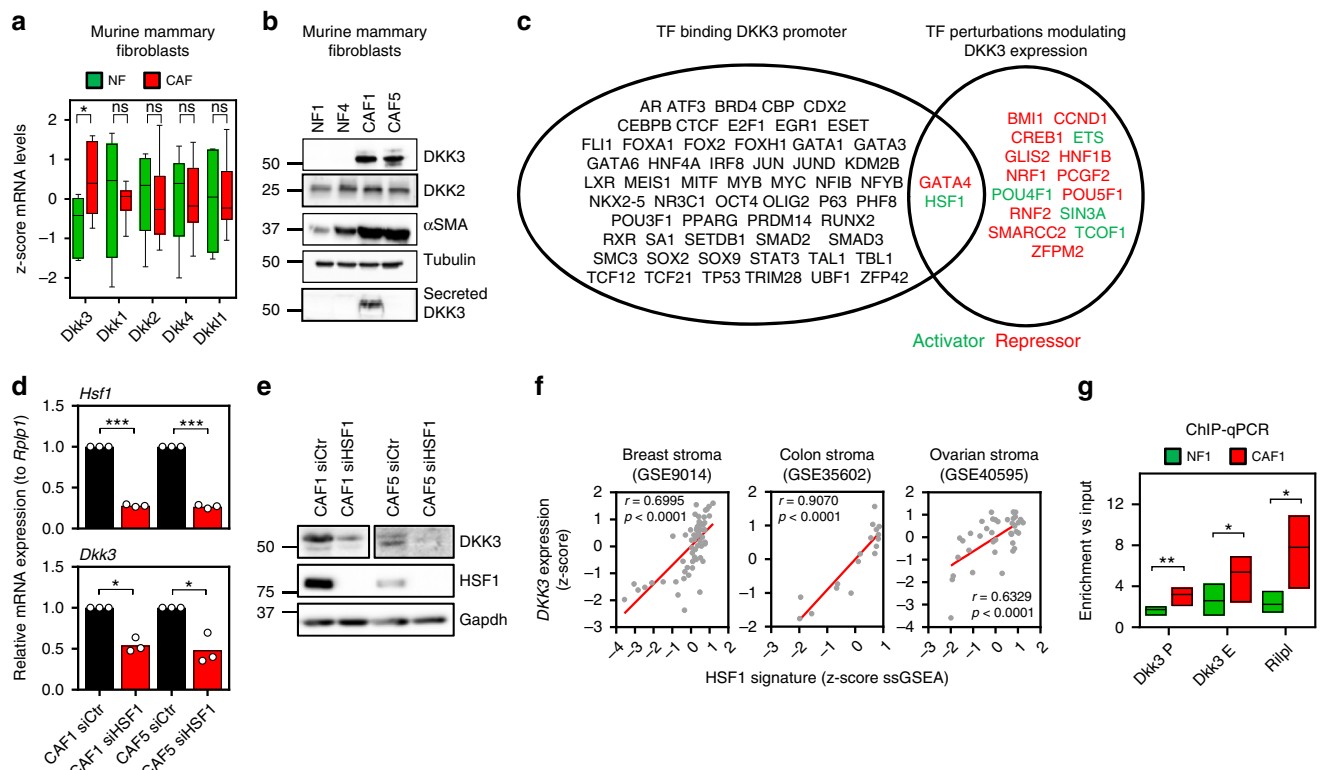

**Fig. 2** DKK3 is a HSF1 target gene associated with CAF emergence. **a** Tukey boxplots showing z-score values of Dickkopft genes mRNA expression in murine mammary NFs and CAFs (NF, $n = 6$; CAF, $n = 8$). **b** Western blot showing protein levels of DKK3, DKK2, αSMA and tubulin in total lysates of two sets of murine mammary NFs and PyMT-CAFs. Levels of secreted DKK3 are also shown. **c** Venn diagram showing the overlap between transcription factors (TFs) known to bind the *DKK3* promoter (left) and TFs whose perturbation modulates *DKK3* expression levels (right). In red, TFs that negatively affect *DKK3* expression; in green, TFs that positively affect *DKK3* expression. **d** Graphs show *Hsf1* and *Dkk3* fold mRNA expression levels (relative to *Rplp1*) in murine PyMT-CAFs (CAF1 and CAF5) after transfection with control or Hsf1 siRNAs (smart-pool). Bars show mean value ± SEM ($n = 3$). **e** Western blots show Hsf1, Dkk3 and Gapdh expression in murine PyMT-CAFs (CAF1 and CAF5) after transfection with control or Hsf1 siRNAs (smart-pool). (**f**) Graphs show correlations between *DKK3* gene expression and HSF1 activity as measured by the expression of HSF1 gene signature (z-score normalised) in stroma of breast, colorectal and ovarian cancers. Pearson correlation coefficient (*r*) is shown. Each dot represents z-score values from individual samples. **g** Graph shows binding of Hsf1 to regulatory elements in the *Dkk3* gene (DKK3 P – promoter; DKK3 E – enhancer) in murine NF1 and CAF1. Primers targeting the bona fide Hsf1 target gene *Rilpl* were used as a positive control. Floating boxes: centre line, mean; box limits, min and max values (normalised to DNA amount)($n = 3$). Where indicated, individual *p* values are shown; alternatively the following symbols were used to describe statistical significance: *$P < 0.05$; **$P < 0.01$; ***$P < 0.001$; #$P < 0.0001$; n.s., non-significant

inactivation and reduced actomyosin contractility, which affects the ability of CAFs to remodel the ECM[7]. *Dkk3* knockdown was associated with reduced phospho-MLC2 and phospho-Src levels (Supplementary Figure 7d). β-catenin siRNA treatment did not affect MLC2 or Src activation (Fig. 6a), suggesting that DKK3 loss-of-function is primarily associated to its effect on YAP/TAZ activity. Confirming this, expression of a constitutively active mutant of YAP in Dkk3-null CAFs was able to recover their gel remodelling and cancer cell growth promoting abilities (Fig. 6e, f). Both DKK3 and YAP gain-of-function phenotypes were associated with reactivation of actomyosin contractility and Src (Fig. 6g). Overall, these data supported a model where HSF1 in CAFs leads to DKK3 upregulation, which in turn potentiates YAP/TAZ activity leading to increased ECM remodelling and promotion of cancer cell growth and invasion (Fig. 6h). In agreement, we observed that silencing Hsf1 in CAFs led to β-catenin and YAP/TAZ inactivation (Fig. 6i and Supplementary Figure 7e) and was associated with a reduction in the matrix remodelling activity of CAFs (Fig. 6j). However, these defects were prevented by constitutive expression of DKK3 or YAP[S5A].

**DKK3 potentiates canonical Wnt signalling in CAFs**. We next investigated the molecular mechanism whereby DKK3 modulates

YAP/TAZ. We observed that the expression levels of key regulators of YAP/TAZ function, MST1/2 and LATS kinases[26], was marginally affected in one of the Dkk3-null clones (i.e. KO.9) when compared to wild-type and KO-REC CAFs (Supplementary Figure 8a). However, analysis of YAP S127 phosphorylation, the main maker of Hippo activity over YAP[26–28], showed no differences after DKK3 modulation. This suggested that DKK3 might be acting independently of Hippo signalling. To confirm this, we silenced LATS1&2 expression in wild type and KO.9 CAFs and assessed CAF functions (Supplementary Figure 8b). Silencing LATS1&2 increased the abilities of CAFs to remodel gels and promote cancer cell growth (Supplementary Figure 8c, d), in line with their negative role in YAP function. However, the presence or absence of DKK3 did not significantly alter the effect of inhibiting Hippo signalling in CAF activities or YAP/TAZ levels, excluding a role for Hippo kinases in YAP/TAZ activation through DKK3.

Recent studies have shown that, similar to β-catenin, YAP/TAZ stability and function are positively modulated in response to canonical Wnt signalling[29,30]. Thus, association of LRP5/6 co-receptors with Axin1 in response to Wnt stimulation displaces β-catenin and YAP/TAZ from the destruction complex. In the absence of LRP5/6 signal, Axin1 targets β-catenin and YAP/TAZ

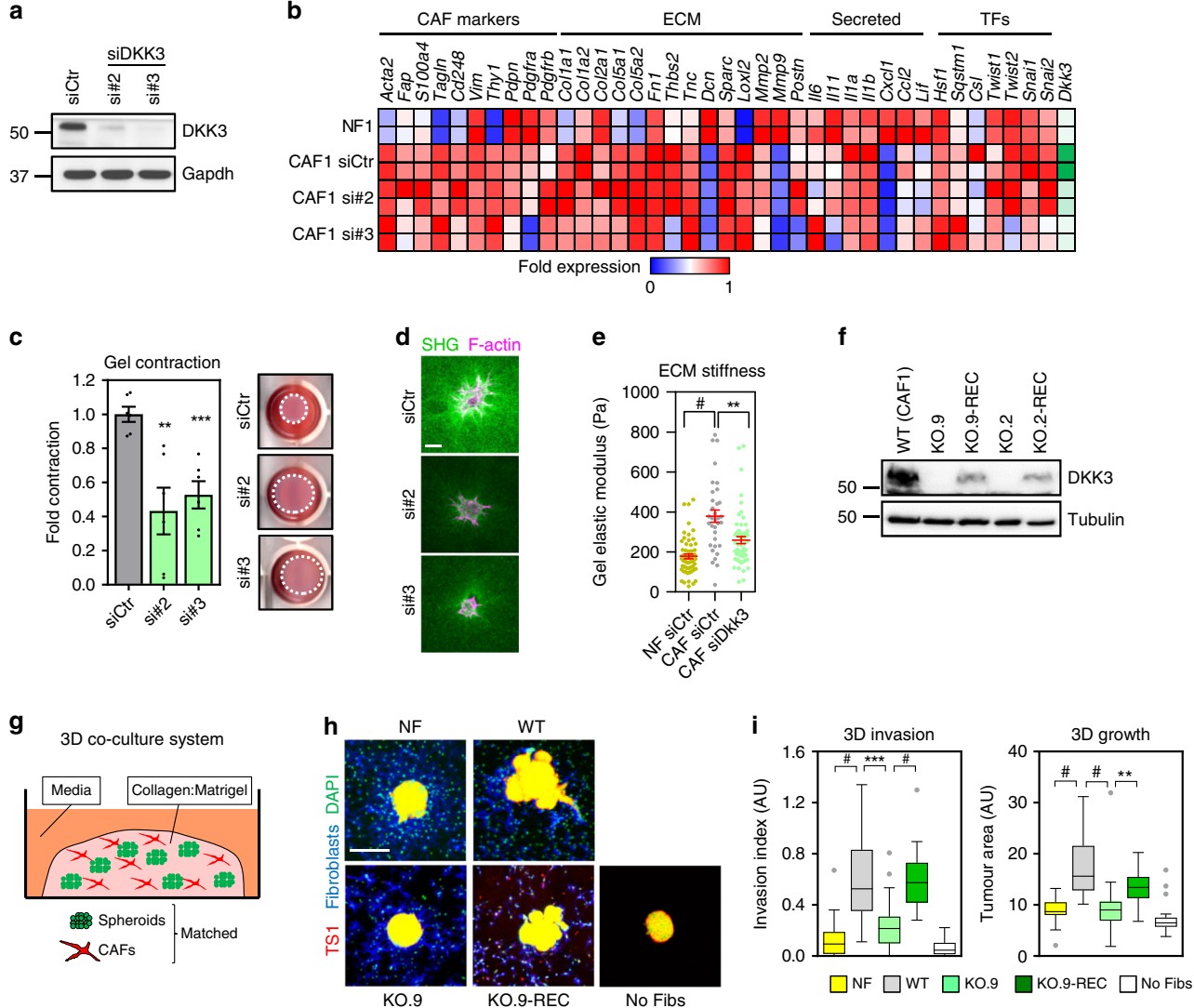

**Fig. 3** DKK3 is a crucial regulator of CAF functions. **a** Western blots showing DKK3 and Gapdh expression in murine CAF1 after transfection with control (siCtr) and 2 independent DKK3 siRNAs (si#2 and si#3). **b** Colour-coded grid showing fold expression of genes generally associated to CAFs in NF4 and wild-type CAF1 after transfection with control and 2 different DKK3 siRNAs. Colours range from red to blue representing, respectively, the lowest (zero) and highest fold activity (one). Genes are grouped into CAF markers, extracellular matrix (ECM) components/remodellers, secreted factors and transcription factors (TFs). Expression for *Dkk3* is shown in a scale of green colour. **c** Histogram shows gel contraction by CAF1 after transfection with control (siCtr) and 2 independent DKK3 siRNAs. Bars represent mean ± SEM (*n* = 6). Images show representative gels remodelled. **d** Images show F-actin (magenta) and collagen second harmonic (SHG, green) in gels remodelled by CAF1 after transfection with control (siCtr) and 2 different DKK3 siRNA. Scale bar, 50 μm. **e** Histogram shows Young's elastic modulus of NF1 or CAF1 cells following transfection with control (siCtr) and DKK3 siRNA (si#sp, smart-pool). Lines represent mean ± SEM (*n* = 34 or more individual measurements). **f** Western blot showing levels of DKK3 and tubulin in wild-type CAF1 (WT) and two sets of DKK3 knockout CAF1 clones (KO) and their recovery counterparts where DKK3 was stably re-expressed (KO-REC). **g** Cartoon describing the experimental set-up for the 3D co-culture system. Cancer cell spheroids are embedded in a collagen:Matrigel matrix containing fibroblasts and fed with media containing 10% FBS for 4–7 days. **h** Images show representative end-point TS1 murine breast cancer spheroids (red) obtained after 3D co-culture with WT, KO.9 and KO.9-REC CAFs (in blue). Spheroids obtained by mono-culture or by co-culture with NFs (in blue) are also shown. DAPI staining (green) was also used. Scale bar, 200 μm. **i** Tukey boxplots show the invasion index (3D invasion) and tumoral area (3D growth) measured from spheroids described in (h); *n* > 26 individual spheroids out of 3 independent experiments. Where indicated, *$P < 0.05$; **$P < 0.01$; ***$P < 0.001$; #$P < 0.0001$; n.s., non-significant

for degradation. Gene expression analyses indicated that modulating DKK3 expression levels did not alter the expression of Wnt signalling regulators (Supplementary Figure 8e). However, our previous data suggested that DKK3 depletion/ectopic expression was affecting β-catenin and YAP/TAZ protein expression (Supplementary Figures 6b&g and Supplementary Figure 8a). To confirm whether DKK3 was affecting the activity of the destruction complex, we performed cycloheximide-chase assays. Fig. 7a shows that the protein stability of β-catenin and

YAP/TAZ was severely affected in the absence of DKK3. Further analyses revealed that LRP6 levels at the cell surface were diminished when *Dkk3* was knocked-out (Fig. 7b), suggesting that DKK3 was affecting YAP/TAZ activity and CAF functions at the level of LRP5/6. In agreement, targeting *Lrp5/6* expression in WT-CAFs (Supplementary Figure 8f), attenuated β-catenin and YAP activation (Fig. 7c and Supplementary Figure 8f) and diminished their tumour-promoting activities (Fig. 7d, e). Yet, silencing *Lrp5/6* expression in Dkk3-null CAFs had no functional

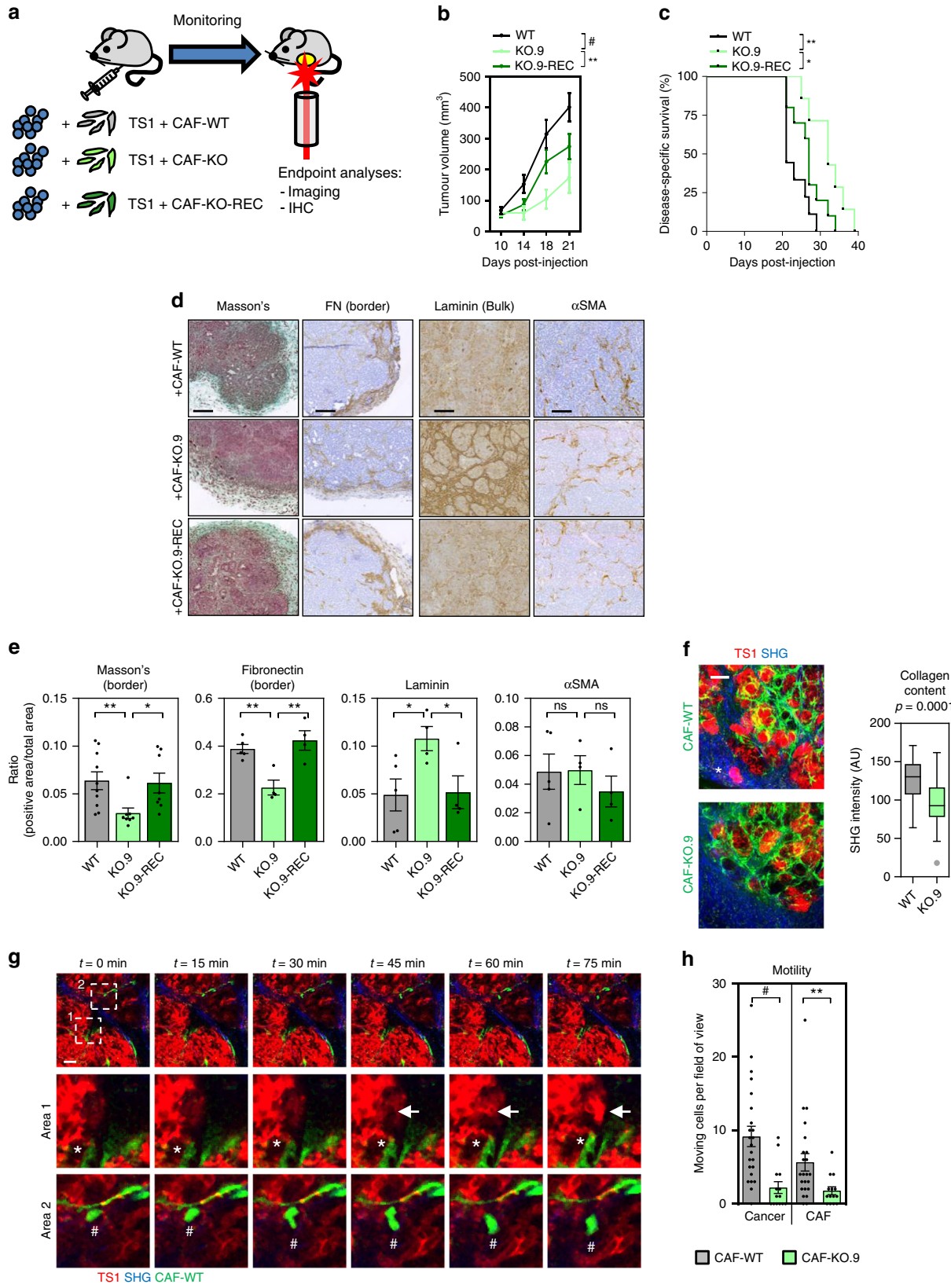

consequences. Inhibiting canonical Wnt signalling with XAV939[31] yielded similar results (Supplementary Figure 7g&h), suggesting that this was a canonical Wnt-related function. To further confirm this, we investigated the role of DKK3 in CAFs after Wnt ligand stimulation. We observed that stimulation of wild type and Dkk3-null CAFs with canonical Wnt ligand Wnt3a

induced a significant increase in their basal (i.e. low serum) gel remodelling activities, whereas non-canonical Wnt5a stimulation had no effect (Fig. 7f). Even though DKK3 was not absolutely required for CAFs to respond to Wnt3a, the gel remodelling activities of CAFs after Wnt3a stimulation were significantly reduced in the absence of DKK3. Importantly, these effects were

**Fig. 4** DKK3 promotes the pro-tumorigenic behaviour of CAFs in vivo. **a** Cartoon describing the experimental set-up used to assess the tumour-promoting potential of CAFs in vivo by subcutaneous co-injection of TS1 murine cancer cells and indicated CAFs. **b** Graph showing the volumes of tumours in syngeneic mice (FVB/n) at the indicated days post-injection. Lines represent mean ± SEM ($n = 7$ or more individual tumours). **c** Survival curves for tumours as described in (b), representing the percentage of animals alive at the indicated time point after injection. Individual $p$ values are shown. **d** Representative images showing Masson's trichrome, fibronectin (FN), laminin and αSMA staining of indicated tumours at day 14 post-injection. Scale bars, 100 μm. **e** Charts show quantification of positive areas relative to total areas for stainings in (d). Bars indicate mean ± SEM ($n = 4$ individual tumours except for WT-CAF tumours that $n = 5$). **f** Representative images of intravital imaging of tumours generated by co-injection of TS1 cancer cells (red) with CAF-WT or CAF-KO.9 CAFs (green) in CD-1 nude mice at day 20. Collagen fibres (blue) were imaged by SHG. Asterisk indicates areas of cancer cell invasion at the tumour border. Scale bars, 100 μm. Tukey boxplot shows collagen content as calculated by SHG intensity, $n = 24$ or more fields of view at the tumour border from 3 independent tumours. **g** Representative time-lapse images from intravital imaging of TS1 (red) and CAF-WT (green) tumours in CD-1 nude mice. Collagen fibres (blue) were imaged by SHG. Zoom-up areas with cancer cell movement associated to a CAF (Area 1, asterisk), cancer cell movement towards a CAF-enriched area (Area 1, arrow) and CAF movement (Area 2, hashtag) are also shown. Scale bar, 50 μm. **h** Graph showing analysis of cancer cell and CAF movement by intravital imaging. Bars indicate average number of moving cells per field of view ± SEM (CAF-WT, $n = 23$ fields of view from 3 independent tumours; CAF-KO, $n = 14$ fields of view from 2 independent tumours). For all graphs, $*P < 0.05$; $**P < 0.01$; $***P < 0.001$; $\#P < 0.0001$; n.s., non-significant

concomitant with β-catenin and YAP activation (Fig. 7g, h), indicating that DKK3 promotes YAP/TAZ signalling by potentiating Wnt signalling in CAFs.

**DKK3 disable the negative Wnt regulator Kremen1&2 in CAFs**. DKK proteins have been shown to inhibit Wnt signalling by triggering LRP5/6 internalisation through formation of a ternary complex with Kremen1/2 receptors[9,10]. This is in striking contrast with our observations on the role of DKK3 in CAFs, and underline the functional differences of DKK proteins reported in other systems[32–34]. DKK3 is an unusual member of the family as it does not interact with LRP5/6 but still can interact with Kremen[9,10,35]. Using HEK293T cells, it has been proposed that DKK3-Kremen interaction negatively affects surface expression of Kremen, which may favour canonical Wnt signalling by still unclear mechanisms[35]. We first confirmed that Kremen1 but not LRP6 immunoprecipitated with DKK3 in CAFs (Fig. 8a). In addition, DKK3-Kremen1 co-localised to internal structures; however, after DKK3 silencing Kremen1 localised to the cell periphery (Fig. 8b), in an opposite pattern to LRP6 localisation (Fig. 8c). Interestingly, DKK3 expression was inversely associated to changes in Kremen1&2 protein expression (Fig. 8d). mRNA levels of *Kremen1/2* were not altered after *Dkk3* silencing (Supplementary Figure 8e), suggesting a post-translational regulation. Cycloheximide-chase assays confirmed that Kremen1 was less stable in WT-CAFs than in Dkk3-null CAFs whereas LRP6 stability was increased (Fig. 8e). Our data suggested that disabling Kremen activity may restore LRP6 functions in the absence of DKK3 expression. In agreement, silencing Kremen receptors in Dkk3-null CAFs (Fig. 8f) increased their abilities to remodel gels (Fig. 8g) and promoted cancer cell growth and invasion (Fig. 8h, i). These changes were associated with concomitant re-localisation of LRP6 to the cell surface (Fig. 8j) and reactivation of YAP/TAZ and β-catenin signalling (Fig. 8f, k). Conversely, knocking-down *Kremen1&2* in DKK3-expressing CAFs had no functional consequences (Supplementary Figure 8i).

Overall, our mechanistic analyses support a model whereby DKK3 stabilises the cell-surface levels of LRP6 by uncoupling LRP6 from the Kremen-mediated internalisation machinery. In turn, DKK3-mediated LRP6 regulation leads to activation of β-catenin and YAP/TAZ, with the latter being the main mediator of DKK3 functions (Fig. 9).

## Discussion

Secreted DKK proteins are emerging as unpredicted tumour-promoting factors due to their immune-suppressive activities[32,34,36]. Here, we demonstrate that the family member

DKK3 can also promote aggressive behaviours in CAFs in a cell autonomous manner. DKK3 reduces YAP/TAZ degradation by potentiating Wnt signalling, thus acting in parallel to YAP/TAZ regulation via mechanotransduction[7,37]. DKK3 also activates β-catenin in CAFs and we document that cancer stroma and CAFs present increased β-catenin activity. However, inhibition of β-catenin by RNAi did not affect CAF-mediated ECM remodelling, cancer cell growth and invasion. Further studies are required to determine if β-catenin modulates other pro-tumorigenic functions in CAFs such as potentiating the tumour initiating capacity of cancer cells or immune suppression[12]. The role of DKK3 as a positive regulator of canonical Wnt is in contrast to the well documented role of other DKK family members. Here, we provide mechanistic insights into these differences. Contrary to what has been shown for DKK1 and DKK2, we show that DKK3 does not bind LRP6 co-receptor in CAFs, and therefore cannot fulfil the bona fide antagonist role of DKK proteins in canonical Wnt signalling. By contrast, we demonstrate that DKK3 perturbs the negative regulator Kremen and enhances Wnt signalling via LRP6.

Whilst several CAF effectors have been identified, an unanswered question is how the coordination of different signalling pathways involved in their control is established. Our results suggest that HSF1-dependent DKK3 upregulation could be a response of stromal fibroblasts to stresses found in the tumour microenvironment. In turn, DKK3 expression promotes fibroblast activation as it sustains crucial activating pathways. Thus, DKK3 expression potentiates canonical Wnt signalling, leading to reduced YAP/TAZ degradation. This leads to YAP/TAZ activation, ECM stiffening and the generation of aggressive tumour microenvironments. We propose that this may be a mechanism for tumour-promoting CAFs to be selected as tumour progresses and a plausible mechanism explaining its significant enrichment in cancer stroma. Further work is still required to understand the processes modulating HSF1 activity in CAFs, and whether the axis HSF1-DKK3-YAP is also relevant in other systems.

To conclude, we identify a key role of DKK3 in tumour stroma. It is required for many of the pro-tumorigenic functions of CAFs, including matrix stiffening and cancer cell growth and invasion. High levels of DKK3 in the stroma are sustained by HSF1, potentiate Wnt signalling and reduce YAP/TAZ degradation. Thus, DKK3 acts as a crucial integrator of key molecular players modulating the emergence of tumour-promoting phenotypes in CAFs.

## Methods

**Mouse strains**. Transgenic FVB/n mice expressing the Polyoma Middle T antigen oncogene under the Mouse Mammary Tumour Virus promoter (MMTV-PyMT)[38]

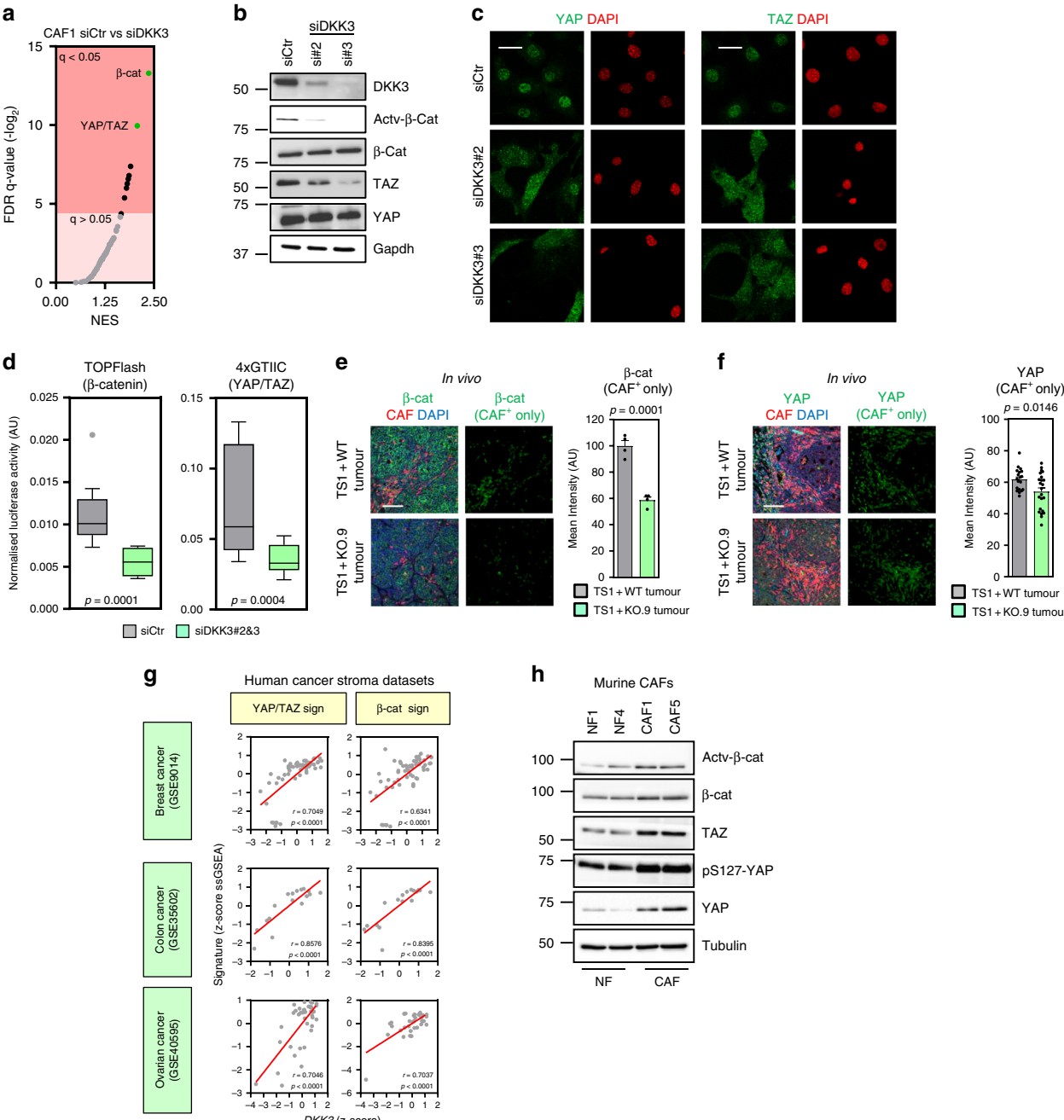

**Fig. 5** DKK3 potentiates β-catenin and YAP/TAZ signalling in CAFs. **a** Graph showing GSEA of murine CAF1 after transfection with control siRNA (CAF-siCtr) vs CAF1 after transfection with DKK3 siRNAs (si#2 and si#3). The graph indicates the Normalised Enrichment Score (NES) and False Discovery Rate (FDR) $q$-value (-$\log_2$) for each gene set. In green, top differential gene-sets, including β-catenin and YAP/TAZ. **b** Western blot showing DKK3, non-phospho (active) β-catenin (Ser33/37/Thr41), β-catenin, TAZ,YAP and Gapdh levels in CAF1 after transfection with control (siCtr) and two independent DKK3 siRNA. **c** Images show YAP (left panels) or TAZ (right panels) localisation (green) and DAPI staining (red) in CAF1 after transfection with control (siCtr) and two independent DKK3 siRNA. Scale bars, 20 μm. **d** Tukey boxplots show luciferase activity (Firefly/Renilla) indicative of β-catenin activation (TOPFlash reporter) or YAP/TAZ activation (4xGTIIC-lux reporter) in CAF1 after transfection with control (siCtr) or DKK3 (si#2&3) siRNAs; $n = 9$ for β-catenin, $n = 17$ for YAP/TAZ. **e** Left panels show total β-catenin (green), S100A4 (CAF marker, red) and DAPI (blue) staining of TS1 tumours admixed with WT or KO.9-CAFs. Right panels show processed images showing β-catenin staining in CAF-positive areas. Scale bar, 100 μm. Chart shows mean β-catenin intensity in CAFs. Bars represent mean ± SEM ($n = 4$ fields of view). **f** Left panels show total YAP (green), S100A4 (CAF marker, red) and DAPI (blue) staining of TS1 tumours admixed with WT or KO.9-CAFs. Right panels show processed images showing YAP staining in CAFs. Scale bar, 100 μm. Chart shows mean YAP intensity in CAFs. Bars represent mean ± SEM ($n = 19$ or more fields of view). **g** Graphs show correlations between *DKK3* gene expression and the expression of CAF-specific YAP/TAZ and β-catenin signatures (z-score normalised) in normal and cancerous stroma from mammary gland, colorectal and ovarian human tissues. Pearson correlation coefficient ($r$) and $p$ values are shown. **h** Representative Western blot showing non-phospho (active) β-catenin (Ser33/37/Thr41), β-catenin, TAZ, phospho-S127-YAP, YAP and tubulin in murine NF1, N4, CAF1 and CAF5. For all graphs, *$P < 0.05$; **$P < 0.01$; ***$P < 0.001$; #$P < 0.0001$; n.s., non-significant

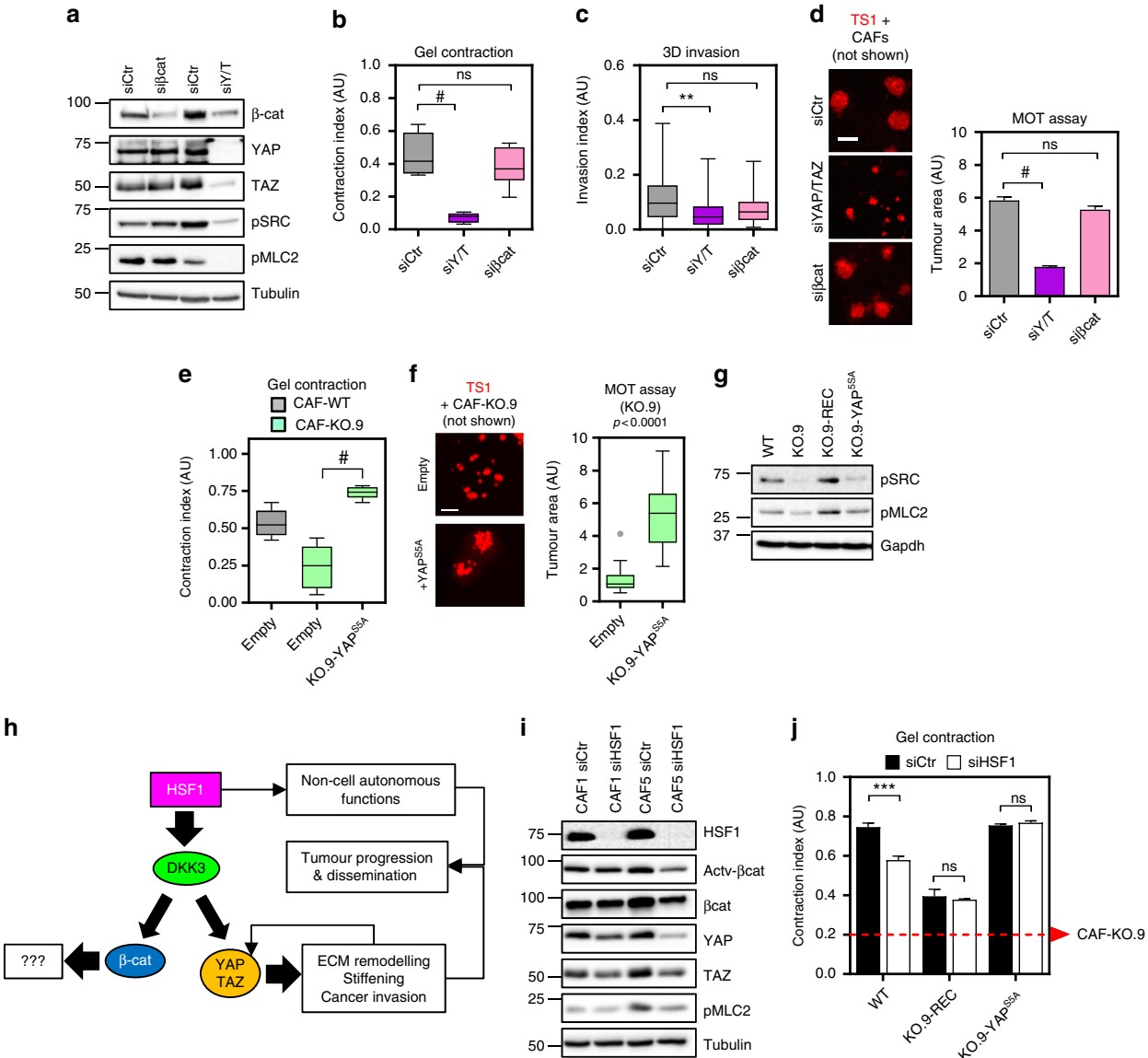

**Fig. 6** DKK3 promotes aggressive behaviours in CAFs via YAP/TAZ. **a** Western blots showing β-catenin, YAP, TAZ, pY416-Src (pSRC), pS19-MLC2 (pMLC2) and Tubulin in CAF1 following transfection with control (siCtr), β-catenin (siβcat; smart-pool) or YAP/TAZ (siY/T; smart-pool) siRNAs. **b** Tukey boxplot shows gel contraction of CAF1 transfected as in (**a**); $n = 8$ gels except for siY/T that $n = 6$. **c** Tukey boxplot shows invasion index (3D invasion) measured from TS1 spheroids obtained after 3D co-culture with CAF1 transfected as in (**a**); $n > 32$ individual spheroids out of 3 independent experiments. **d** Representative images of end-point TS1 (red) MOT co-culture with CAF1 transfected as in (**a**). Scale bar, 100 μm. Graph shows the area of colonies. Bars represent mean ± SEM ($n = 570$ or more colonies from at least 3 independent experiments). **e** Tukey boxplot shows gel contraction index of CAF-WT or CAF-KO.9 stably expressing empty vector (empty) or constitutive active YAP mutant (YAP[S5A]); $n = 7$ gels or more. **f** Representative images of end-point TS1 (red) MOT co-culture with CAF-KO.9 stably expressing empty vector (empty) or YAP[S5A]. Scale bar, 100 μm. Tukey boxplot shows the area of colonies; $n = 28$ or more colonies from at least 3 independent experiments. **g** Western blot showing levels of pY416-Src (pSRC), pS19-MLC2 (pMLC2) and Gapdh in CAF-WT or CAF-KO.9 stably expressing empty vector (KO.9), DKK3 (KO.9-REC), or YAP[S5A]. **h** Schematic diagram showing the model integrating our findings. HSF1 in CAFs upregulates DKK3 which in turn potentiates YAP/TAZ and β-catenin signalling. YAP promotes actomyosin contractility leading to ECM remodelling and cancer cell growth and invasion. The role of β-catenin is still undetermined. **i** Western blots show Hsf1, non-phospho (active) β-catenin (Ser33/37/Thr41), β-catenin, YAP, TAZ, pS19-MLC2 (pMLC2) and Tubulin in murine CAF1 and CAF5 after transfection with control (siCtr) or Hsf1 (smart-pool) siRNAs. **j** Graph shows gel contraction of CAF-WT, CAF-KO.9-REC and CAF-KO.9-YAP[S5A] after transfection with control (siCtr) or Hsf1 (smart-pool) siRNAs. Bars represent mean ± SEM ($n = 3$ or more). A dashed red line indicates the average contraction by KO.9 CAFs. For all graphs, *$P < 0.05$; **$P < 0.01$; ***$P < 0.001$; #$P < 0.0001$; n.s., non-significant

were used for tumour cell isolation and FACS analysis. Wild-type FVB/n and CD-1 Nude mice (Charles River) were used as recipients for tumours. All animals were kept in accordance with UK regulations under project license PPL80/2368.

**Cell lines**. Established murine fibroblasts from FVB/n normal mammary glands (NFs) and MMTV-PyMT mammary carcinoma (CAFs) have been previously

described[7,39]. Human fibroblasts from normal mammary glands and breast carcinomas were provided by Julia Tchou (University of Pennsilvania, USA)[40] and Clare Isacke (Institute of Cancer Research, UK). Human fibroblasts from rectal carcinomas (RC11), colon carcinomas (CAF25) and adjacent normal tissue (NAD-D) were provided by Danijela Vignjevic (Institute Curie, France). Human fibroblasts from ovarian carcinoma (EOC.TIL.04) were provided by Marco Donia (University of Copenhagen, Denmark). Human CAFs from cervical (Cer-CAF)

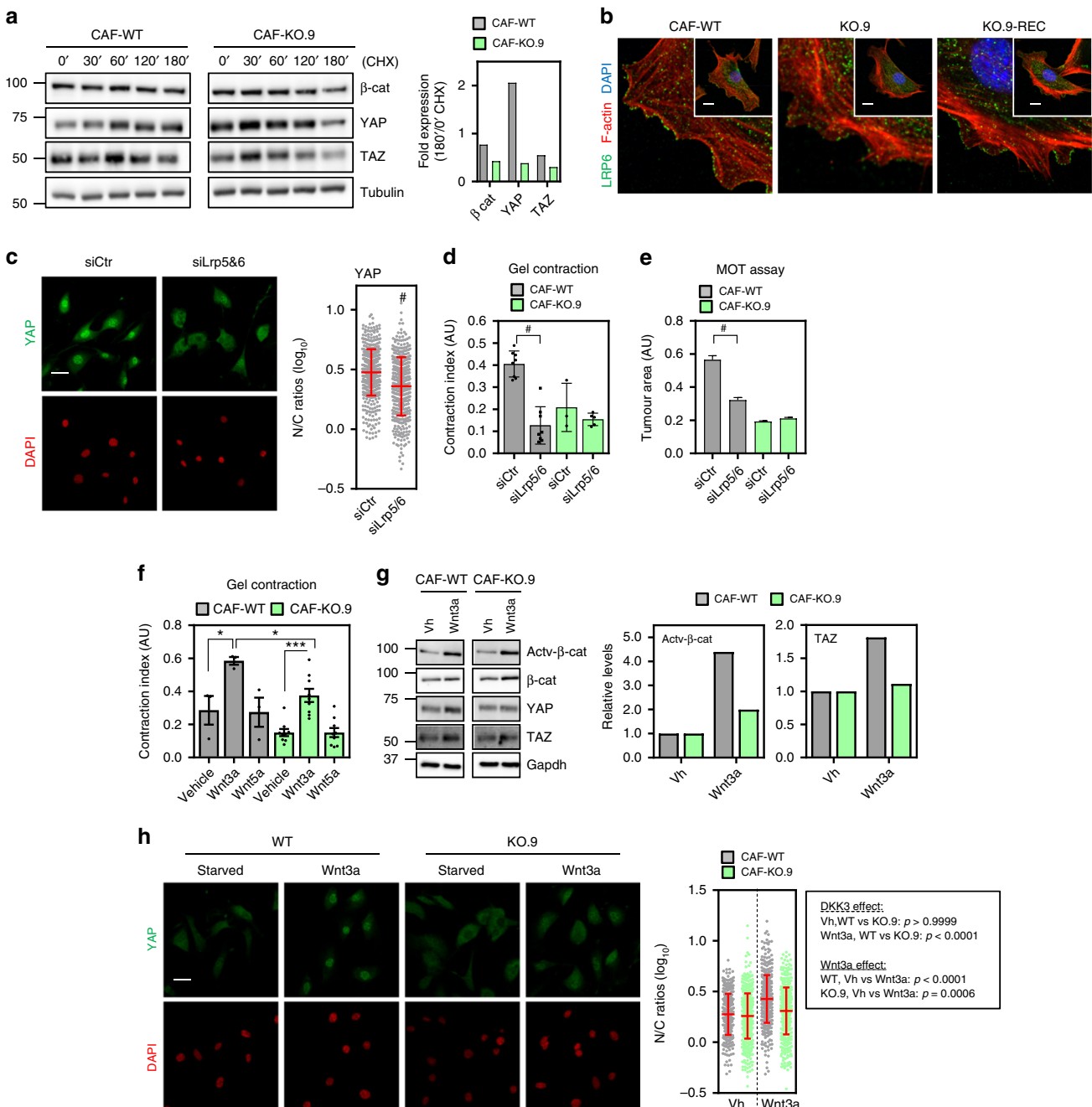

**Fig. 7** DKK3 potentiates canonical Wnt signalling and affects YAP/β-catenin degradation. **a** Western blots show β-catenin, TAZ, YAP and tubulin expression in murine WT and KO.9 CAFs after treatment with 100 μg mL$^{-1}$ cycloheximide (CHX) for the indicated times (in min). Graph represents quantification of the indicated blots (normalised to tubulin) at 180 min relative to the amount at 0 min. **b** Low and high magnification images of LRP6 (green), F-actin (red) and DAPI (blue) staining of murine WT, KO.9 and KO.9-REC CAFs in 1% FBS. Scale bars, 15 μm. **c** Images show YAP (green) and DAPI (red) staining of CAF1 after transfection with control (siCtr) and LRP5&6 siRNA (smart-pool). Bar, 50 μm. Graph shows quantification of nuclear relative to cytosolic fluorescent intensity (log$_{10}$ ratios) of YAP. Lines represent mean ± SEM. **d** Graph shows gel contraction index of murine WT and KO.9 CAFs after transfection with control (siCtr) or Lrp6 (siLrp5/6) siRNAs. Bars represent mean ± SEM ($n = 3$ or more). **e** Graph shows the area of colonies formed by TS1 MOT co-culture with WT and KO.9 CAFs after transfection with control (siCtr) or Lrp5&6 (siLrp5/6) siRNAs. Bars represent mean ± SEM ($n = 584$ or more colonies from at least 3 independent experiments). **f** Graph shows gel contraction of WT and KO CAFs cultured in 5% FBS and stimulated with vehicle or 200 ng mL$^{-1}$ of Wnt3a or Wnt5a. Bars represent mean ± SEM ($n = 3$). KO data are merged results of KO.2, KO.7 and KO.9 on triplicate ($n = 9$). **g** Western blot showing levels of non-phospho (active) β-catenin (Ser33/37/Thr41), β-catenin, YAP, TAZ and Gapdh in WT and KO.9 CAFs after stimulation with vehicle (−) or 200 ng mL$^{-1}$ of Wnt3a for 3 h. Graph represents quantification of blots indicating the fold levels normalised to Gapdh. **h** Images show YAP (green) and DAPI (red) staining of murine WT and KO.9 CAFs after 16 h starvation followed by stimulation with vehicle (Starvation) or 200 ng mL$^{-1}$ of Wnt3a for 3 h. Bar, 50 μm. Graph shows quantification of nuclear relative to cytosolic fluorescent intensity (log$_{10}$ ratios) of YAP. Lines represent mean ± SEM. Individual $p$ values for different comparisons are shown. For all graphs, *$P < 0.05$; **$P < 0.01$; ***$P < 0.001$; #$P < 0.0001$; n.s., non-significant

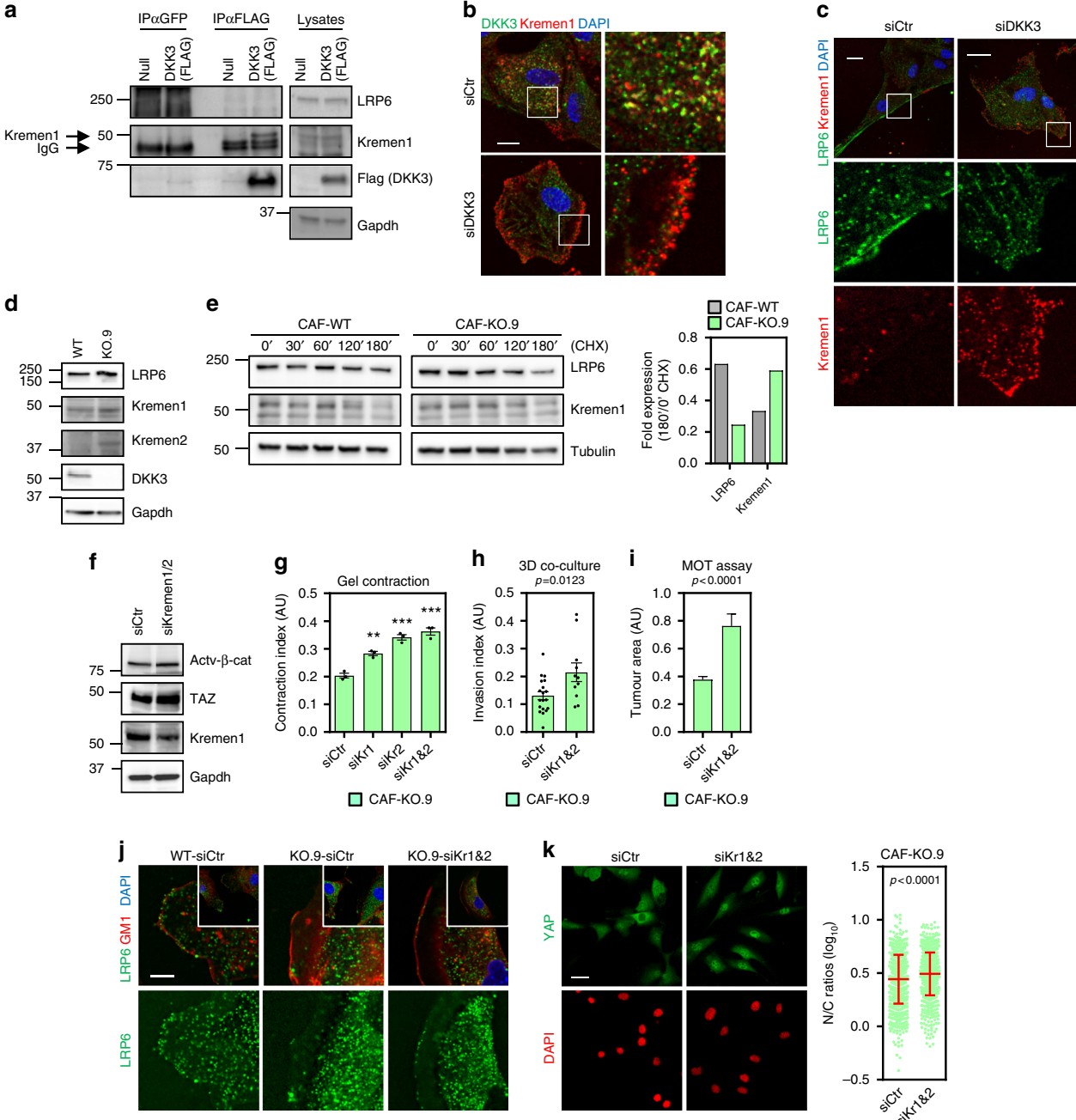

**Fig. 8** DKK3 regulates the balance between LRP6 and Kremen and affects YAP/TAZ signalling. **a** Western blots show anti-Flag co-immune-precipitation of LRP6 and Kremen1 in human BC-CAFs (TB147) expressing Flag-DKK3 or empty vector (null). **b** Images show DKK3 (green), Kremen1 (red) and DAPI (blue) staining of human BC-CAF TB165 after transfection with control (siCtr) and DKK3 siRNA (smart-pool). Scale bars, 15 μm. **c** Images show LRP6 (green), Kremen1 (red) and DAPI (blue) staining of human BC-CAF TB165 transfected as in (**b**). Scale bars, 15 μm. **d** Western blots show LRP6, Kremen1, Kremen2, DKK3 and Gapdh expression in WT and KO.9 CAFs. **e** Western blots show LRP6, Kremen1 and tubulin expression in indicated CAFs after treatment with 100 μg mL$^{-1}$ cyclohexamide (CHX) at different times (in min). Graph represents quantification of the indicated blots (normalised to tubulin) at 180 min relative to the amount at 0 min. **f** Western blots show non-phospho (active) β-catenin (Ser33/37/Thr41), TAZ, Kremen1 and Tubulin CAF-KO.9 after transfection with control (siCtr) and Kremen1&2 siRNA (smart-pool). **g** Graph shows gel contraction of KO.9 CAFs after transfection with control (siCtr), Kremen1 (siKr1), Kremen2 (siKr2) or Kremen1&2 (siKr1&2) siRNAs (smart-pool). Bars represent mean ± SEM ($n = 4$ or more individual gels). **h** Graph shows the invasion index of TS1 spheroids after 3D co-culture with KO.9 CAFs transfected as in (**f**). Bars represent mean ± SEM ($n = 11$ or more individual spheroids, 2 independent experiments). **i** Graph shows the tumoral area of TS1 MOT co-culture with KO.9 CAFs transfected as in (**f**). Bars represent mean ± SEM ($n = 348$ or more colonies, at least 3 independent experiments). **j** Images show LRP6 (green), GM1 (red) and DAPI (blue) staining of indicated CAFs. Cells were fixed and subjected to staining without permeabilisation. Scale bar, 15 μm. **k** Images show YAP (green) and DAPI (red) staining of KO.9 CAFs transfected as in (**f**). Bar, 30 μm. Graph shows quantification of nuclear relative to cytosolic fluorescent intensity (log$_{10}$ ratios) of YAP. Lines represent mean ± SEM. For all graphs, *$P < 0.05$; **$P < 0.01$; ***$P < 0.001$; #$P < 0.0001$; n.s., non-significant

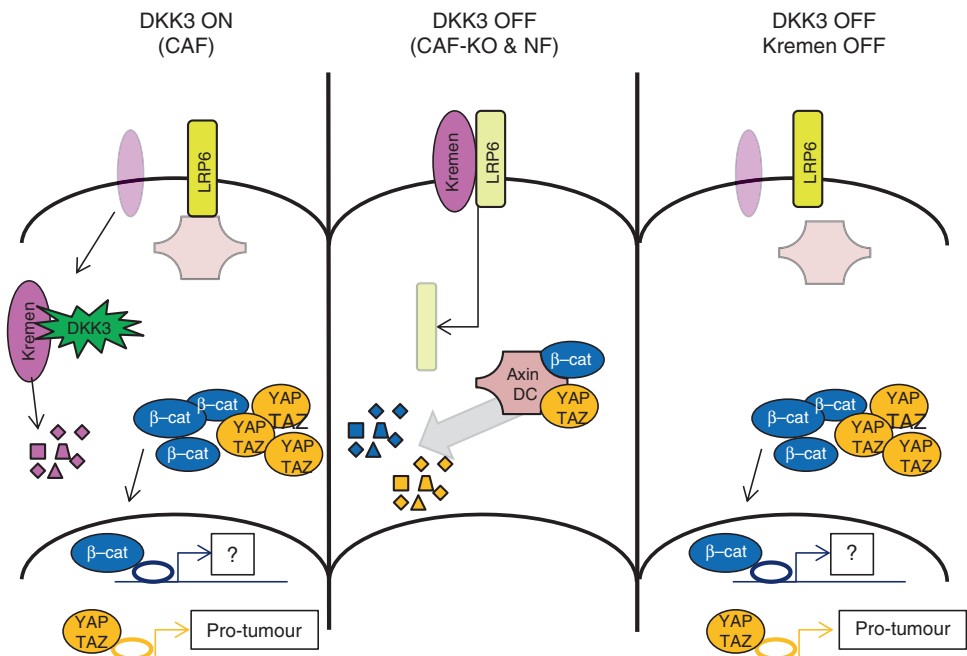

**Fig. 9** Model depicting the proposed mechanism of DKK3-mediated regulation of YAP/TAZ and β-catenin in CAFs. On CAFs, DKK3 destabilises Wnt negative regulator Kremen leading to increased LRP6 membrane localisation, which in turn stabilises YAP/TAZ and β-catenin levels via canonical Wnt. Whereas β-catenin signalling is dispensable for CAFs to remodel the ECM and promote cancer cell growth and invasion, DKK3-driven YAP activation is required to induce a tumour-promoting phenotype. Absence of DKK3 in DKK3-null CAFs and NFs is associated with decreased YAP/TAZ and β-catenin activity. In CAFs, loss of DKK3 leads to concomitant upregulation of Kremen, LRP6 inactivation and YAP/TAZ and β-catenin destabilization. In this scenario, depletion of Kremen1/2 is able to rescue LRP6 membrane localisation and YAP/TAZ and β-catenin activity

squamous cell carcinoma have been previously described[7]. Human primary fibroblasts were expanded and immortalised using hTERT virus (pCSII vector backbone) followed by selection with hygromycin. All resulting fibroblast populations were assessed for fibroblast and CAF marker expression and thoroughly characterised[7]. All fibroblasts were cultured in DMEM (Sigma), GlutaMax (Gibco), 10% FBS, 1% insulin-selenium-transferrin (ITS, Gibco). MMTV-PyMT TS1 murine breast cancer cells were used in most assays and to generate tumours in syngeneic FVB/n mice. Human breast cancer cell line BT20 was cultured in MEM plus GlutaMAX and 10% FBS. Colon cancer cell line HCT116 was grown in McCoy 5 A (with 1% GlutaMAX and 10% FCS). Ovarian cancer cell line SKOV3 was grown in RPMI with 1% GlutaMAX, 10% FBS and 1% Pyruvate. All cell lines were grown in a standard humidified 5% CO2 incubator at 37 °C and tested negative for mycoplasma infection with MycoAlert™ (Lonza). CAFs and cancer cell lines were fluorescently labelled using the following lentiviral vectors, as indicated: EGFP-CAAX pCSII-IRES2-hygro, ECFP-CAAX pCSII-IRES2-hygro, ORANGE-CAAX PCSII-IRES2-hygro, mCherry-CAAX pCSII-IRES2-hygro and pCSII-IRES2-blasti-eGFP.

**cDNA, RNAi and reagents.** DKK3 (a kind gift by Peter Berger, University of Innsbruck) was ectopically expressed in CAF/NFs lines using a pLENTI6 lentiviral vector system. shRNAs targeting human DKK3 were cloned into the lentiviral vector pLKO.1-TRC-cloning vector (Addgene, Plasmid #10878) at AgeI/EcoRI restriction sites according to the online Addgene protocol (www.addgene.org/tools/protocols/plko). shRNA sequences were obtained from the GPP Web Portal (Broad Institute, http://portals.broadinstitute.org/gpp/public/). A list of the respective shRNA sequences can be found in Supplementary Table 3. siRNAs were purchased from Dharmacon and are listed in the Supplementary Table 4. For in vitro treatments the following growth factors and drugs were used: recombinant human DKK3 (R&D, 1118-DK-050), XAV939 (Sigma, X3004), Cycloheximide (Sigma, C7698), Wnt3a (R&D, 5036-WN-010) and Wnt5a (R&D, 645-WN-010). Lentiviral plasmid for expression of mutant YAP was 5SA-YAP1-YFP vector (kind gift of Eric Sahai, Crick Institute).

**Generation of CRISPR knockout cell lines.** The CRISPR plasmid U6-gRNA/CMV-Cas9-GFP containing a guide RNA targeting mouse DKK3 was purchased from Sigma (MM0000346166). CAF1 cells (CAF-WT) were transfected with the plasmid using Lipofectamine (Life Technologies) following manufacturer's instructions and GFP positive cells were single sorted into 96 well plates after 24 h. GFP-negative clones were also sorted as controls (e.g. WT.9). Individual cell clones were expanded and the DKK3 locus targeted by CRISPR was sequenced for

knockout validation. In addition, loss of DKK3 protein expression was also confirmed by immunoblotting to generate DKK3-null CAFs (CAF-KO). DKK3-null clones were infected with DKK3-expressing lentivirus to generate KO-CAFs re-expressing DKK3 (CAF-KO-REC).

**Conditioned media.** CAF-derived conditioned media was generated by culturing a confluent monolayer of cells in DMEM 2.5% or 5% FBS for 48 h. Media was then recovered and filtered through a 0.22 μm filter and used as indicated. For detection of DKK3 in the conditioned media, 50 μL of StrataClean resin (Agilent Technologies UK Ltd, 400714) were added to 1 mL of media. Each sample was vortexed for 1 min, left at RT for 1 min and spun 15,000× g for 5 min at 4 °C. The supernatant was then removed and the beads were prepared in protein sample buffer to be resolved in SDS-PAGE.

**Transfections.** Fibroblasts were seeded at 60% confluency and transfected using DharmaFECT 1 (Dharmacon) for siRNA (100 nM final concentration), and Lipofectamine (Life Technologies) for plasmids following manufacturer's instructions. Cell lines stably expressing cDNA (DKK3 or GFP/CFP/mCherry) or shRNAs were generated by lentiviral infection followed by puromycin selection for 2 weeks (2 μg mL⁻¹). Alternatively, fluorescent-labelled cells were sorted by FACS.

**ECM-remodelling assay.** To assess force-mediated matrix remodelling, $7.5 \times 10^4$ murine fibroblasts or $3–5 \times 10^4$ human fibroblasts were embedded in 120 μL of a mixture of collagen I (BD Biosciences) and ECM gel mixture (Sigma) yielding a final collagen concentration of ~4.6 mg mL⁻¹ and a final ECM gel mix concentration of ~2.2 mg mL⁻¹ (Collagen-rich matrix hereafter). Once the gel was set, cells were maintained in fibroblasts medium. Gel contraction was monitored daily by scanning the plates. Unless stated otherwise, the gel contraction value refers to the contraction observed after 2 days. To obtain the gel contraction value, the relative diameter of the well and the gel were measured using ImageJ software, and the percentage of contraction was calculated using the formula 100 × (well area − gel area) / well area. When indicated, gels were fixed in 4% paraformaldehyde (PFA) for 16 h and processed for immunofluorescence. Backscattered light (reflectance) was collected to image the matrix surrounding cells. Alternatively, collagen second-harmonic generation (SHG) signal was acquired by exciting with an 880 nm pulsed Ti–Sapphire laser and acquiring emitted light at 440 nm using a Leica SP8 microscope.

**3D co-culture.** The 3D co-culture spheroid assay was adapted from Dolznig et al[19]. In short, cancer cells were seeded in low numbers (200 cells per well TS1-mCherry, 50 cells per well HCT116-mCherry, 150 cells per well BT20-GFP and SKOV3-orange) in ultra-low adherent cell culture plates (Corning) to form spheroids for 48 h. The spheroids were collected and mixed with $1 \times 10^5$ fibroblasts. The spheroid and fibroblast suspension was centrifuged at $200 \times g$ for 4 min and the supernatant was carefully removed. The pellet was first re-suspended in 30 μL of medium and then mixed with 270 μL of Collagen-rich matrix. The 300 μL gel containing the spheroids and fibroblasts was seeded in a glass-bottom 3.5 cm MatTek dish. A nylon filter (Nylon NET filters 120 μm, Merck Millipore) with a hole of ~1 diameter cut out at the centre was added to prevent excessive gel contraction. The gel was set for 1 h in the incubator at 37 ˚C and cancer cell medium was added. The spheroids were incubated for 5–7 days (3–5 days for studies involving siRNA-transfected CAFs) and then fixed with 4% PFA and either imaged directly or stained additionally with DAPI and phalloidin-TRITC or phalloidin-FITC. Samples were mounted and analysed using a Leica SP8 confocal microscope. Confocal sections were acquired for individual spheroids from bottom to top, z-stack projections were analysed using ImageJ and data analysed in GraphPad Prism. The invasion index was calculated by measuring the total area over which cancer cells had dispersed (including invading and non-invading cells) and the area of non-invading cells (1 – [non-invading area/total area]). To calculate *tumour growth index*, area covered by cancer cells in each spheroid was calculated.

**Matrigel on Top (MOT) co-culture.** 24-well MatTek glass bottom plates were pre-coated with 100 μL of Matrigel. Next, $6 \times 10^4$ CAFs per well were seeded, unless otherwise stated. After 1 h, fluorescently labelled cancer cells ($2 \times 10^4$ per well) were seeded and allowed to adhere for 4 h. Growth medium was exchanged to DMEM 2.5% FCS and 2% Matrigel and co-cultures were allowed to grow for 48 h, or up to 120 h for the ovarian cells. Co-cultures were then fixed in 4% PFA and analysed by confocal microscopy. Images were analysed using ImageJ and data analysed in GraphPad Prism. To calculate cancer cell growth index, area covered by cancer cells in each field of view was calculated.

**Atomic force microscopy.** To assess the elastic modulus of gels remodelled by fibroblasts, $75 \times 10^3$ fibroblasts were embedded in 100 μL of Collagen-rich matrx and seeded on ultra-low attachment 96-well plate (Costar). Once the gels were set, they were maintained in fibroblast medium. After three days, the elastic modulus of the gels was measured as previously described[7]. Gels were gently lifted from their well and fixed in the centre of 50 mm glass bottom Petri dishes using cyanoacrylate superglue. Once glued, Leibovitz L-15 medium (Invitrogen) supplemented with 10% FCS was added to the dish. AFM measurements were performed with a JPK Nanowizard-I (JPK instruments, Berlin, Germany) interfaced to an inverted optical microscope (IX-81, Olympus). AFM cantilevers with pyramidal tips (MLCT, Bruker, Karlsruhe, Germany) and nominal spring constants of 0.07 Nm⁻¹ were modified by gluing 35-μm radius glass beads to the cantilever underside with UV curing glue (UV curing, Loctite, UK). Cantilever spring constants were determined prior to modification using the thermal noise method implemented in the AFM software (JPK SPM). Prior to any indentation tests, the sensitivity of the cantilever was set by measuring the slope of force-distance curves acquired on glass regions of the Petri dish. Using the optical microscope, the tip of the cantilever was aligned over regions in the middle of the gel and, for each gel, measurements were acquired in 30–40 locations ~100 μm apart. Force-distance curves were acquired with an approach speed of 5 μm.s⁻¹ until reaching the maximum set force of 3 nN. After the experiment, the elastic moduli were extracted from the force-distance curves by fitting the contact portion of curves to a Hertz contact model. For each force-distance curve, goodness of fit was evaluated by calculating r^2 values and only fits with r^2 > 0.80 were retained for further analyses (representing on average 80% of the acquired force-curves).

**Immunofluorescence.** Cells were fixed in 4% PFA for 1 h and permeabilised by incubation in PBS 0.2% Triton 100 (Sigma) at RT for 20 min. For HSF1 immunofluorescence, cells were permeabilised by incubation in PBS 0.5% NP-40 (Sigma) at 4 °C for 20 min (twice), in PBS 0.3% Triton 100 (Sigma) at RT for 20 min and in PBS 0.1% Triton 100 at RT for 15 min (twice). Where indicated (non-permeabilised), this step was not performed. Samples were blocked for 60 min at RT in blocking solution: 4% BSA PBS 0.05% Tween20 (Sigma). Then, cells were incubated with primary antibody in blocking solution in a wet chamber overnight at 4 °C. After three washes of 15 min in PBS, secondary antibody in blocking solution was added for 3 h. After three washes of 15 min in PBS, the samples were mounted and analysed using a Leica SP8 confocal microscope. To quantify β-catenin activation by immunofluorescence, for each field of view we calculated the area positive for active-β-catenin staining per cell using Volocity. These values were then normalised to the mean value of CAF-WT (murine CAFs) or control siRNA-transfected cells (human CAFs). To quantify YAP and TAZ activation by immunofluorescence, we measured their nuclear localisation using Volocity. Briefly, we calculated for each cell the total intensity of nuclear HSF1, YAP or TAZ staining (determined by the intensity within the region delimited by DAPI staining) and perinuclear staining (defined as a region encompassing two to seven pixels from the nucleus border). Nuclear/cytoplasmic YAP and TAZ ratios were calculated as the

log₁₀ of the total nuclear intensity/mean perinuclear region intensity per cell. For the analysis of human CAFs, nuclear/cytoplasmic YAP and TAZ ratios were normalised to the mean ratio of control siRNA-transfected cells. Antibody description and working dilutions can be found in Supplementary Table 2.

**Tissue immunohistochemistry and immunofluorescence.** For histological analysis, tumours were dissected into 10% neutral buffered formalin, embedded in paraffin blocks and serial sections were taken. Paraffin-embedded tissue sections were rehydrated before antigen retrieval using pH 6 sodium citrate buffer. After blocking endogenous peroxidase (DAKO peroxidase block), sections were incubated for 1 h at RT with primary antibodies. For DKK3 staining, four antibodies (Abgent AP1523a, R&D AF1118, Sigma HPA011868 and Santa-Cruz H130) were tested for specificity using western blot and cell pellets in control and DKK3-silenced CAFs, and then optimised for TMA analysis. Both Sigma and Santa-Cruz antibodies yielded satisfactory results. Data presented correspond to Santa-Cruz DKK3 antibodies. Sections were incubated with secondary antibodies for 50 min at RT, treated with 3,30-diaminobenzidine and counterstained with haematoxylin. Images were acquired using a Digital Slide Scanner (Hamamatsu Photonics). Images were analysed using Velocity and data analysed in GraphPad Prism. For each staining, positive area was calculated and normalised to the total area for all slices processed. For immunofluorescence, after antigen retrieval sections were incubated with 0.2% Triton X-100 (Sigma) 10% Goat serum (Sigma) blocking solution for 1 h RT. Sections were incubated overnight at 4 °C with primary antibodies diluted in PBS. After washing, the sections were incubated with fluorescent-conjugated secondary antibodies (AlexaFluor) diluted in PBS for 1 h at RT. Confocal images were captured using a Leica SP8 microscope. Quantitative analysis of YAP and β-catenin staining in the tumour stroma was performed using Volocity software. Briefly, fibroblasts were identified using automated threshold based on S100A4 staining. The mean intensities of YAP and β-catenin were then measured. In addition to haematoxylin and eosin staining and Masson's trichrome staining, antibody description and working dilutions can be found in Supplementary Table 2.

**Tissue microarrays.** The following tissue microarrays were purchased from US Biomax, Inc: breast cancer (BR1504b), colon cancer (CO2083) and ovarian cancer (OV2001a). Formalin-fixed, paraffin-embedded (FFPE) sections were deparaffinized, blocked with 3% H₂O₂ and antigen retrieval was performed using a pressure cooker with Dako citrate buffer (pH 6.0). Slides were incubated with DKK3 antibody (DKK-3 H-130, Santa Cruz, sc-25518) for 1 h at RT. Visualisation was achieved with 3,30-diaminobenzidine as a chromogen (Dako Envision + System). Images were acquired using a Digital Slide Scanner (Hamamatsu Photonics). DKK3 staining was quantified using the weighted Histoscore method to give a value of 0–300[41].

**Tissue digestion for cell isolation and FACS analysis.** MMTV-PyMT tumours were minced with a scalpel and digested with a mixture of DNase and Liberase (Roche Diagnostics). On enzymatic digestion, samples were passed through a 100-μm filter. Cells were incubated for 5 min at RT in 2 mL NH₄Cl solution (0.8% in H₂O) to eliminate red blood cells. Cells were then directly used for FACS staining and sorted with a FACSAria flow cytometer (BD Biosciences). Single cell suspensions of tumour cells were labelled with FITC Rat Anti-Mouse CD45 Clone 30-F11 (BD Pharmingen™, 553079), APC Rat Anti-Mouse CD31 Clone MEC 13.3 (BD Pharmingen™, 561814), PE/Cy7 anti-mouse CD326 (Ep-CAM) (BioLegend, 118215) and PE anti-mouse CD140a (PDGFRa) (BioLegend, 135905). For cell sorting, four populations (CD45⁺ for immune cells, CD31⁺ for endothelial, EPCAM⁺ for epithelial and PDGFRA⁺ for CAFs) were collected and processed for RNA extraction and qRT-PCR.

**Proliferation analysis.** For proliferation analyses with conditioned media, TS1 cancer cells were seeded at 5,000 per well in a 24-well plate in duplicates and incubated with media containing 5% FBS (vehicle) or 5% FBS plus 100 ng mL⁻¹ recombinant DKK3 (rDKK3), or with CAF conditioned medium (5% FBS) for 72 h. Where indicated, blocking antibody for DKK3 (R&D AF1118) or isotype control (R&D AB-108-C) were added at 1 μg mL⁻¹. AlamarBlue® assay was used according to manufacturer's instructions to compare cell growth and viability.

**Scratch Wound healing assay.** TS1 cancer cells were cultured to a confluent monolayer in 24-well plates. A straight scratch was made on the cell layer using a 200 μL tip across the centre of the well. Cell debris was removed by washing twice with PBS and cells were incubated with media containing 5% FBS (vehicle) or 5% FBS plus 100 ng mL⁻¹ recombinant DKK3 (rDKK3), or with CAF conditioned medium (5% FBS). Where indicated, blocking antibody for DKK3 (R&D AF1118) or isotype control (R&D AB-108-C) were added at 1 μg mL⁻¹. Cell migration was monitored for 40 h using live cell imaging in the IncuCyte® (Essen Bioscience) in a standard humidified 5% CO2 incubator at 37 °C. The wounded area at start and end of experiment was measured using ImageJ. Wound closure was calculated as the percentage of healed area (area at $t_0$ minus area at $t_{40}$) to the starting wounded area.

**Co-Immunoprecipitation.** For Kremen1 and DKK3 co-inmunoprecipitations, CAFs expressing empty vector of Flag-DKK3 were lysed in immunoprecipitation lysis buffer (50 mM Tris-HCl pH 7.5, 150 mM NaCl, 1% Triton-X-100, 10% glycerol, 2 mM EDTA, 25 mM NaF and 2 mM NaH2PO4) containing protease and phosphatase inhibitor. Lysates were pre-cleared using G-sepharose, incubated with anti-Flag antibody (rabbit) or IgG immobilised on G-sepharose beads at 4 °C overnight. Beads were then washed five times with lysis buffer and eluted with 20 μL of 2x SDS sample buffer.

**Western blotting.** Protein lysates and immunoprecipitants were processed following standard procedures. Enhanced-chemiluminescence signal was acquired using an Azure Biosystems c600. Exposures within the dynamic range were quantified by densitometry using ImageJ. Antibody description and working dilutions can be found in Supplementary Table 2. Immunoblot images in Figures and Supplementary Figures show molecular weight markers in kDa. Supplementary Figure 9 shows the uncropped images corresponding to the panels displayed in the main figures.

**Luciferase reporter assays.** The following plasmids were used for luciferase reporter assays: TEAD-reporter construct (8xGTIIC-luciferase, Addgene, Plasmid #34615), TOP-reporter construct (M50 Super 8x TOPFlash, Addgene, Plasmid #12456) and CMV-Renilla (pGL4.75[hRluc/CMV], Promega). CAFs were seeded in 6-wells plates at 70% confluency. Cells were transfected with 2 μg TOPFlash or TEAD-reporter and 100 ng of CMV-Renilla cDNA constructs using Lipofectamine 3000 according to the manufacturer's instructions. Cells were lysed 48 h after transfection in 100 μL of lysis buffer (Promega). Aliquots of the cell lysates were used to read luciferase emission using Dual-Glo® Luciferase Assay System (Promega) according to the manufacturer's instructions. Luciferase activities were normalised to Renilla activity.

**Generation of syngeneic orthotopic tumours.** MMTV-PyMT TS1 murine breast cancer cells were used to generate tumours in wild-type FVB/n. Briefly, $10^6$ TS1 cells and $3 \times 10^6$ CAF (WT, KO.9 or KO.9-REC) were suspended in 100 μL of PBS:Matrigel (50:50) and injected subcutaneously into 6–8 week old females. Tumour size was measured every other day using callipers. To calculate tumour volume the formula $V = (length \times width^2)/2$ was used. Mice were killed once tumours reach the maximum allowed size.

**Intravital imaging.** The implantation of an orthotopic mammary imaging window in CD-1 nude mice was performed as previously described[42]. Forty-eight hours after surgical implantation of the imaging window, a mixture of $1 \times 10^6$ CAFs (GFP) and $4 \times 10^5$ TS1 (CAAX-mCherry) cancer cells in a PBS:Matrigel solution (50:50) were injected under the window. Intravital 2-photon imaging was performed at specified time-points using a Leica SP8 microscope. Briefly, mice were anaesthetised and tumours were excited with an 880 nm pulsed Ti–Sapphire laser and emitted light acquired at 440 nm (collagen second-harmonic generation, SHG) and 530 nm (GFP). In addition, a mCherry signal was acquired sequentially using confocal microscopy. During ~10-min intervals, 5 to 8 different regions were imaged simultaneously for 2 h for each tumour. In each region, a z-stack of 4–5 images (approximately 50 μm deep on average) was taken at a spacing of 10 μm between images, resulting in a time-lapse three-dimensional z-series for analysis. To analyse the *collagen content*, mean intensity of SHG signal of the top confocal plane for each z-stack was used. Moving cells were defined as those that moved 10 μm or more during the length of each movie.

**RNA isolation and qRT-PCR.** To obtain RNA from the different fibroblast populations, RNA was isolated using EZNA Total RNA Kit 1 (Omega Bio-tek). Reverse transcription was performed using Precision NanoScript 2 Reverse-Transcription-kit (PrimerDesign) and qPCR using PrecisionPLUS 2x qPCR MasterMix with ROX and SybrGreen (PrimerDesign). Expression levels of indicated genes were normalised to the expression of Gapdh, Rplp1 or Lamc2. Sequences of the oligonucleotides used for qRT-PCR are described in the Supplementary Table 5. For array analysis, RNA was isolated as described and processed in collaboration with Eurofins Genomics (GeneChip Mouse Gene 2.0 ST Array—details available on reasonable request).

**ChIP-qPCR.** $7–10 \times 10^6$ NF or CAF at subconfluency were fixed in 1% formaldehyde and sonicated using the Bioruptor Pico sonication device (Diagenode; B01060001) using 15 cycles (30 s on; 30 s off) at maximum intensity. Purified chromatin was then separated for: (i) immunoprecipitation using Dynabeads Protein G (Life Technologies: 10003D) coated with 8 μg and 4 μg of HSF1 antibody (ThermoFisher: RT-405-P, and Cell Signalling: 4356 S, respectively) per ChIP experiment; (ii) non-immunoprecipitated chromatin, used as Input control; and (iii) assessment of sonication efficiencies using a 1% agarose gel. For the quantitative PCR briefly, reactions were carried out in 10 μL volume containing 5 μL of Sybergreen mix (Applied Biosystems; 4472918), 0.5 μL of primer (5 μM final concentration), 2.5 μL of genomic DNA and 2 μL of DNAse/RNAse-free water. A three-step cycle programme and a melting analysis were applied. The cycling steps

were as follows: 10 s at 95 °C, 30 s at 60 °C and 30 s at 72 °C, for 40 cycles. Enrichment of the immunoprecipitated sample was confirmed using positive and negative controls. The exact loci of the primers are as follows: negative control (gene desert): chr6:116908976–116909177; *Dkk3* Promoter (P): chr7:112159485–112159638; *Dkk3* Enhancer (E): chr7:112183116–112183344; *Rilpl*, within the gene (positive control): chr5:124510011–124510190.

**Gene expression analyses of clinical datasets.** Gene expression analyses of human tumour stroma were retrieved from NCBI Gene Expression Omnibus (GEO). Datasets are described in Supplementary Figure 1a and include: Finak (GSE9014, Breast), Karnoub (GSE8977, Breast), Yeoung (GSE40595, Ovary), Nishida (GSE35602, Colon), Saadi (GSE19632, Aesophagus), Costea (GSE38517, Oral SCC), Navab (GSE22863, Lung) and Sherman (GSE43770, Pancreas). Additional datasets include: GSE14548 (Breast cancer epithelia and stroma), GSE20086 (human breast NF and CAFs), GSE70468 (human colon NF and CAFs) and GSE35250 (human ovarian NF and CAFs). In addition, the dataset describing the expression profiles of cell populations purified from human colorectal cancer (GSE39396) was also used. For Affimetrix-based arrays, probe-to-gene mapping was performed using Jetset[43]; for the rest of the arrays, highest variance probes were selected. Probes mapping *DKK* genes used for each array can be found in Supplementary Figure 1a. Unless otherwise stated, expression values for each gene were z-score normalised. For the analyses of YAP/TAZ and β-catenin signatures in tumour stromal datasets, we first generated CAF-specific YAP/TAZ and β-catenin signatures. Genes significantly downregulated in murine CAF1 after transfection with YAP/TAZ siRNA or XAV939 (β-catenin inhibitor) treatment were identified. In addition, we sought for genes specifically expressed by FAP-positive CAFs in human tumours using the GSE39396 dataset. This dataset describes the expression profiles of cancer cells, immune cells, endothelial cells and CAFs purified from human colorectal cancer. We identified genes significantly upregulated more than 1.5-fold in FAP-positive samples when compared to the rest of the samples. Genes from the YAP/TAZ and β-catenin signatures that overlapped with the genes in the *FAP-positive signature* were selected into *CAF-specific* signatures; the rest were selected into *CAF-unspecific* signatures. To calculate the gene-signature score in each sample, we used single sample Gene Set Enrichment Analysis (ssGSEA)[44] Projection Software from the GenePattern platform developed by the Broad Institute of MIT and Harvard (USA) and available at GenePattern (www.genepattern.broadinstitute.org), following the programs guidelines (log normalisation; weighting exponent 0.75). This analysis calculates separate enrichment scores for each pairing of sample and gene set. Each ssGSEA enrichment score represents the degree to which the genes in a particular gene set are co-ordinately up- or downregulated within a sample. ssGSEA scores were z-score normalised. For analyses of HSF1 signature, a similar approach was performed. The top 200 genes downregulated between wild-type and Hsf1-null mouse embryonic fibroblasts (GSE56252) were identified and genes also present in the *FAP-positive signature* were selected to generate the HSF1 *CAF-specific* signature.

**Enrich database analysis.** To identify putative transcription factors (TFs) regulating *DKK3* expression we used the Enrich database[14,15] [http://amp.pharm.mssm.edu/Enrichr/]. We queried DKK3 gene and retrieved the *ChEA 2016* dataset to identify possible TFs that bind the DKK3 promoter, obtaining a list of 58 TFs (Supplementary Data 1). In addition, we retrieved the *TF-LOF Expression* from GEO dataset to obtain a list of TFs whose loss of function has been associated to changes in DKK3 expression (Supplementary Table 6).

**Survival analyses.** Analysis of clinical relevance of *DKK3* expression was assessed using publicly available data from the Kaplan–Meir Plotter platform for breast and ovarian cancer[45,46] (version 2014 and 2015, respectively), as well as the GSE17538 dataset for colorectal cancer[47]. Probe-to-gene mapping was performed using Jetset[43]. For survival analysis of ovarian cancer (progression-free survival), the highest quartile of gene expression was used to dichotomise the different tumours into high and low groups. Survival analyses for breast (recurrence-free survival) and colon (disease-specific survival) datasets, tumours were split into high and low based on median expression, respectively. For breast cancer, further analyses of recurrence-free survival and distant-metastasis-free survival were performed based on ER-status and molecular subtype. Breast cancer disease-free survival analysis based on *DKK3* expression in the tumour microenvironment was performed using the Finak Dataset[48]. Only ER-negative patients were analysed. Mean *DKK3* expression was used to split tumours into high and low expression groups.

**Gene-set enrichment analyses (GSEA).** Array data were processed and analysed using the Gene-set enrichment analysis software, developed by the Broad Institute of MIT and Harvard (USA) and available at www.broadinstitute.org, following the programme guidelines. The specific settings applied in all analyses are: Number of Permutations (1000), Permutation Type (Gene set), Enrichment statistic (Weighted), and Metric for ranking genes (t Test). Values represent the False Discovery Rate $(-\log_2)$ and the Nominal Enrichment Score *(NES)* of each gene set. The list of the specific gene sets analysed and their sources are available in Supplementary Data 2.

**Statistical analyses**. Statistical analyses were performed using GraphPad Prism (GraphPad Software, Inc.). When $n$ permitted, values were tested for Gaussian distribution using the D'Agostino-Pearson normality test. For Gaussian distributions, paired or unpaired two-tailed Student's $t$-test and one-way ANOVA with Tukey post-test (for multiple comparisons) were performed. For non-Gaussian distributions, Mann–Whitney's test and Kruskal–Wallis test with Dunn's post-test (for multiple comparisons) were performed. Unless stated otherwise, mean values and standard errors (SEM) are shown. Survival curves were estimated based on the Kaplan–Meier method and compared using a log-rank test. $P$ values of less than 0.05 were considered statistically significant. Where indicated, individual $p$ values are shown; alternatively the following symbols were used to describe statistical significance: $*P < 0.05$; $**P < 0.01$; $***P < 0.001$; $\#P < 0.0001$; n.s., non-significant.

**Reporting Summary**. Further information on experimental design is available in the Nature Research Reporting Summary linked to this article.

## Data availability

Gene expression datasets of NF control or overexpressing DKK3, and CAF1 after transfection with control or DKK3 siRNAs are available at the NCBI GEO under GSE114056 [https://www.ncbi.nlm.nih.gov/geo/query/acc.cgi?acc = GSE114056]. The rest of the datasets generated and/or analysed during the current study are available from the corresponding author upon reasonable request.

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

## Acknowledgements

We thank Clare Isacke, Marjan Iravani and Sarah Tang in the Breast Cancer Now Research Centre at the Institute of Cancer Research for kindly providing human breast CAFs, and Danijela Vignjevic at Institute Curie for providing human colon CAFs. In addition, we thank Sebastian Guettler, Peter Berger, Robert Kypta, Christoff Niehrs and Erik Sahai for providing us with plasmids; Afshan McCarthy for providing MMTV-PyMT tumours; Johanna Joyce, Rachael Natrajan, Pascal Meier, Udai Banerji and

Richard Houlston for providing cancer cell lines; Jacco van Rheenen for training on mammary window imaging; Frances Daley and Anne Lowe for help with immunohistochemistry; Fredrik Wallberg and members of the FACS/Light microscopy Unit for assistance; members of the Biological Services Unit for help with mouse experiments; and Anguraj Sadanandam for assistance with bioinformatics analyses. We also thank lab members for help and advice throughout this work. This work was funded by Worldwide Cancer Research (Grant 15–0273) and the Institute of Cancer Research. F.C. is also funded by Cancer Research UK (C57744/A22057) and the Ramon y Cajal Research Program (MINECO, RYC-2016–20352). Spanish funding to F.C. is partially supported by the European Regional Development Fund.

## Author contributions

F.C. conceived the study. F.C. and N.F. designed the experiments. N.F., R.R. and F.C. performed and analysed the experiments, with help from I.C., A.J.F., M.L. and M.S., N.D. S. and L.M. performed ChIP analyses. E.M. performed atomic force microscopy analyses. M.C.W.W. and J.T. isolated human CAF lines from ovarian and breast cancer, respectively. F.C. wrote the manuscript. All authors critically read the manuscript and provided intellectual input.

## Additional information

**Competing interests:** The authors declare no competing interests.

**Journal Peer Review Information:** *Nature Communications* thanks the anonymous reviewers for their contributions to the peer review of this work. Peer Reviewer Reports are available.

