## [Peer Review File · Nature Communications]

Reviewers' comments:

Reviewer #1 (Remarks to the Author):

The present manuscript by Ferrari et al describes DKK-3 as a new target for HSF-1 driving the activation of YAP/TAZ in cancer-associated fibroblasts. The experiments show that DKK3-expressing CAFs promote extracellular matrix remodeling and contribute to increased tumor growth and motility. The study is well conducted and brings novel and relevant data indicating that stromal DKK-3 is an important factor in the regulation of tumor progression in different types of cancer.

DKK proteins are well-known inhibitors of Wnt signaling, but here the authors present evidence that DKK3 acts as a facilitator factor of Wnt pathway activation. Ferrari and colleagues show that DKK3 binds and redirects the Kremen receptor out of the plasma membrane. The Kremen proteins are known for promoting the internalization of Wnt coreceptors, LRP5/6. In that way, Wnt signaling is triggered and responsible for increasing YAP/TAZ protein stability. YAP/TAZ are transcriptional coactivators that can be activated by two different signaling pathways, Hippo and Wnt pathways. The authors suggest that DKK3 would be promoting YAP/TAZ activation exclusively through the Wnt pathway. However, based on the presented data additional experiments are required to clearly show whether DDK3-driven increase of YAP/TAZ goes solely through Wnt pathway and there is no role for Hippo kinases in that context. Below we provide specific comments in no order of importance.

Major comments:

- Based on Figure S2n, the authors conclude that the changes induced by stromal DKK3 expression in cancer cells does not occur due to secreted DKK3. However, it seems there is a difference in proliferation and wound healing when looking at the TS1 tumor cells treated with CM from WT and KO.9 CAFs (p value is not shown). To better address whether secreted DKK3 is important for the observed phenotypes, TS1 tumor cells could be incubated with WT CAFs CM in the presence or absence of anti-DKK3. Alternatively, the authors might use the CM from the CAF cell line that does not secrete DKK3 (CAF5) in comparison with the CM coming from CAF1 (CAF cell line capable of secreting DKK3).

- Figure S2l, the graph shows the mean of only two independent experiments and no p value is shown for this result.

- Based on Figure S1a, the authors state that changes in stromal expression of other DKK proteins besides DKK3 is not consistent. However, stromal DKK2 expression also seems to be consistently increased among different types of cancer (p value is <0.05). The change in DKK2 does not negate the possible importance of DKK3 but it is overstated and the authors should discuss this change and what it might mean.

- In figure S4a, the authors assessed the status of Hippo kinases on DKK-3 recovered CAFs by western blot. The authors conclude there is no activation of Hippo kinases when DKK3 expression is

recovered in CAFs. However, LATS/MST1 expression seems to be increased upon DKK3 recovery. In order to exclude a role for Hippo kinases in YAP/TAZ activation through DKK3 in the present context, Hippo kinases should be silenced in KO.9 and KO.9 REC cells. In this setting, YAP/TAZ protein levels, collagen contraction and tumor growth should be analyzed. Alternatively, the authors might consider using the CAF1 cell line treated with siCtrl and siDKK3 instead of KO.9 and KO.9 REC cells.

- DKK proteins are well known to act as inhibitors of Wnt signaling. Here, the authors show mostly through β -catenin activation that DKK3 acts in favor of Wnt signaling in CAFs. However, β -catenin can be activated by alternative pathways, like PI3K-Akt and MEK-MAPK. Considering that, a similar setting as requested on the above topic (using KO.9 and KO.9 REC cells) should be used to assess the YAP/TAZ levels when treating CAFs with recombinant Wnt and/or the XAV939 inhibitor. In that way, the authors would bring more substantial evidence that DKK3 acts as facilitator of the Wnt pathway in CAFs. The figures supporting DKK3's role in YAP/TAZ activation through the Wnt pathway should be moved to the main text and not left as supplementary.

Minor comments:

- There are missing supplementary tables; we could not find them?

-In Figure 2b, typing error – “CAF si#3” should be “CAF1 si#3”.

-In Figure 3, the letter “o” is referred as letter “p” in the respective legend of the figure.

- In Figure S4a, typing error – GAPDH is typed as “GAPADH”

Reviewer #2 (Remarks to the Author):

This new MS by Calvo and coworkers represents a very interesting advancement on our understanding of cancer-associated fibroblasts (CAF). Briefly, the authors found that CAF, but not normal fibroblasts, secrete Dickkopf3 (Dkk3), and that stromal Dkk3 expression correlates with tumor malignancy. The authors also found that Dkk3 is required for tumor malignancy in xenotransplant assays, favoring invasive and proliferative abilities of cancer cells. Mechanistically, the authors propose that Dkk3 activates the Wnt pathway by reducing membrane localization of kremen, an inhibitor of the Wnt-coreceptor LRP5/6. In turn, this would allow the activation of beta-catenin, and the non-canonical activation of the Hippo pathway transducers YAP and TAZ. Only YAP/TAZ, but not beta-catenin, are shown to be important for the pro-malignant activities of Dkk3.

Although the study is well-conducted and most of the data are quite convincing, there are some issues that need to be clarified/emended before publication:

- 1) Dkk3 depletion by siRNAs (Fig. 3b) has stronger effects on beta-catenin and on TAZ than the full Dkk3 knockout (Fig. S4a). Please explain this discrepancy, or show better blots for Fig. S4a.
- 2) To convincingly show that the Hippo pathway is not affected by Dkk3 knockout, please reload the blots of phosphor-YAP in Fig. S4a, normalizing by YAP expression; in alternative, please provide quantification of the corresponding bands, normalized by the expression of YAP.
- 3) The Western blot of Fig. S4c is of poor quality, and should be reloaded. In any case, the reported differences in beta-catenin and TAZ levels are not apparent from these images. Quantifications, normalized to GAPDH, are not adding confidence in this case. It would be more convincing if the authors would be able to present immunofluorescence pictures representative of the data depicted in Fig. S4d, showing that LRP6 knockdown causes nuclear exclusion of YAP.
- 4) The authors must make freely available the microarray gene expression data from normal fibroblasts, normal fibroblasts plus Dkk3, and CAF, either wild-type or Dkk3-KO, that they used in the MS.

We were very pleased with the overall positive nature of the reviewers' comments: including the work *'represents a very interesting advancement on our understanding of cancer-associated fibroblasts (CAFs)'* (reviewer #2) and reviewer #1 stating that *'The study is well conducted and brings novel and relevant data indicating that stromal DKK-3 is an important factor in the regulation of tumor progression in different types of cancer'*.

We also acknowledge that the reviewers have suggested some areas for improvement and are grateful for their thoughtful advice. As a result, we have made major improvements in this revised manuscript. The most significant of these are:

- Confirmation that secreted DKK3 has no effect on cancer cells
- Additional data and discussion on stromal DKK2 vs DKK3
- Analysis of DKK3 in regards to Hippo signaling
- Complementary analyses supporting a role of DKK3 as potentiator of Wnt signaling in CAFs.

We describe the changes in this revised manuscript in a detailed point-by-point rebuttal letter. To assist in the revision, changes in the manuscript have been highlighted in yellow. In addition, to comply with Nature Research Editorial Policy we have introduced changes in some graphs to represent data distribution (i.e. dot-plots or box-and-whisker plots).

Response to Reviewer #1:

The present manuscript by Ferrari et al describes DKK-3 as a new target for HSF-1 driving the activation of YAP/TAZ in cancer-associated fibroblasts. The experiments show that DKK3-expressing CAFs promote extracellular matrix remodeling and contribute to increased tumor growth and motility. The study is well conducted and brings novel and relevant data indicating that stromal DKK-3 is an important factor in the regulation of tumor progression in different types of cancer.

DKK proteins are well-known inhibitors of Wnt signaling, but here the authors present evidence that DKK3 acts as a facilitator factor of Wnt pathway activation. Ferrari and colleagues show that DKK3 binds and redirects the Kremen receptor out of the plasma membrane. The Kremen proteins are known for promoting the internalization of Wnt coreceptors, LRP5/6. In that way, Wnt signaling is triggered and responsible for increasing YAP/TAZ protein stability. YAP/TAZ are transcriptional coactivators that can be activated by two different signaling pathways, Hippo and Wnt pathways. The authors suggest that DKK3 would be promoting YAP/TAZ activation exclusively through the Wnt pathway. However, based on the presented data additional experiments are required to clearly show whether DDK3-driven increase of YAP/TAZ goes solely through Wnt pathway and there is no role for Hippo kinases in that context. Below we provide specific comments in no order of importance.

Major comments:

- Based on Figure S2n, the authors conclude that the changes induced by stromal DKK3 expression in cancer cells does not occur due to secreted DKK3. However, it seems there is a difference in proliferation and wound healing when looking at the TS1 tumor cells treated with CM from WT and KO.9 CAFs (p value is not shown). To better address whether

secreted DKK3 is important for the observed phenotypes, TS1 tumor cells could be incubated with WT CAFs CM in the presence or absence of anti-DKK3. Alternatively, the authors might use the CM from the CAF cell line that does not secrete DKK3 (CAF5) in comparison with the CM coming from CAF1 (CAF cell line capable of secreting DKK3).

We thank the reviewer for this major comment. We apologize for inadvertently missing to include the statistical analyses between TS1 tumour cells treated with CM from WT and KO.9 CAFs in both proliferation and wound healing. Now we provide these analyses showing that there is no statistical difference (One-way ANOVA non-parametric Kruskal-Wallis test) as shown in **Figure S2n**. For the reviewer, we include here the adjusted p values:

Proliferation:

WT-CM vs KO.9 CM: $p=0.2986$

KO.9 CM vs KO.9-REC CM: $p=0.4863$

Wound Healing:

WT-CM vs KO.9 CM: $p=0.0908$

KO.9 CM vs KO.9-REC CM: $p>0.9999$

In addition, following Reviewer#1's suggestions we have investigated the effect of blocking secreted DKK3 function over cancer cell proliferation and motility using a blocking antibody for DKK3 (R&D AF1118, 1 $\mu\text{g}/\text{mL}$ final concentration), and differences between conditioned media from CAF1 (with secreted DKK3) and CAF5 (no detectable secreted DKK3). Our results indicate that there is no significant difference in cancer cell proliferation or motility when cells were treated with conditioned media from CAF1 or CAF5, or when both conditioned media were treated with αDKK3 (**Figure S2o**). We now discuss these additional data in Page 3, Lines 121-124.

- Figure S2l, the graph shows the mean of only two independent experiments and no p value is shown for this result.

We have now repeated this experiment ($n=5$ individual gels) and included bars and statistical analysis, confirming that stable expression of DKK3 in NF significantly increases their gel remodelling capabilities (**Figure S2l**).

- Based on Figure S1a, the authors state that changes in stromal expression of other DKK proteins besides DKK3 is not consistent. However, stromal DKK2 expression also seems to be consistently increased among different types of cancer (p value is <0.05). The change in DKK2 does not negate the possible importance of DKK3 but it is overstated and the authors should discuss this change and what it might mean.

Reviewer#1 is right in that DKK2 presents also a similar trend towards upregulation in the tumour stroma compared to normal stroma, albeit there is one study in breast cancer that indicates the opposite (Karnoub GSE8977) and there are no significant differences in Lung and Pancreas. There were additional data (not included in the original manuscript due to space restrictions) that influenced our decision to focus our study on DKK3 in CAFs, though. These data have now been included in the manuscript. First, we observe that DKK2 expression in the stromal compartment was not restricted to CAFs and presented a wider distribution (**Figure S1c&d**). Second, DKK2 stromal expression did not significantly

associate with survival in ER-negative BC, colon or ovarian cancer patients (**Figure S1f**). Finally, we showed in the original manuscript that there were no differences in DKK2 mRNA or protein expression between NF and CAFs in our cellular models (**Figure 1h&i and S1h**).

These data suggest respectively that: (i) DKK2 upregulation in the stromal compartment may result from other cell types and is not restricted to CAFs; (ii) there is no clinical evidence supporting a role for stromal DKK2 in human disease; and (iii) DKK2 expression is not associated with functional activation of CAFs when compared to NFs. Together, these data did not support an extensive characterisation of DKK2 function in CAFs. We now discuss these additional data in Page 3, Lines 82-87.

- In figure S4a, the authors assessed the status of Hippo kinases on DKK-3 recovered CAFs by western blot. The authors conclude there is no activation of Hippo kinases when DKK3 expression is recovered in CAFs. However, LATS/MST1 expression seems to be increased upon DKK3 recovery. In order to exclude a role for Hippo kinases in YAP/TAZ activation through DKK3 in the present context, Hippo kinases should be silenced in KO.9 and KO.9 REC cells. In this setting, YAP/TAZ protein levels, collagen contraction and tumor growth should be analyzed. Alternatively, the authors might consider using the CAF1 cell line treated with siCtrl and siDKK3 instead of KO.9 and KO.9 REC cells.

We thank Reviewer#1 for this insightful comment. Albeit very marginal, there is a minor downregulation of both LATS and MST1 expression in DKK3-KO CAFs compared to WT and KO-REC. We did not think this was particularly relevant for three reasons: First, this specific phenotype was only evident in one of the KO clones (KO#9) and not in the other (KO#2), suggesting it may be a clone-specific event. We now provide quantification of MST and LATS levels normalised to GAPDH (**Figure S4a**). Second, the levels of pS127-YAP (normalised to total YAP levels) indicate that there were no differences in the S127 phosphorylation after modulating DKK3 expression. Following Reviewer#2's suggestion (see *below*), we now provide quantification for this in **Figure S4a**. S127 phosphorylation is the main marker used in the field for Hippo activity over YAP (Zhao et al, Genes Dev 24, 862-74; Zhao et al, Genes Dev 21, 2747-61; Yu et al Cell 150, 780-91), and was suggested by Reviewer#2 as a good readout of Hippo activity over YAP in our system. Thus, even though LATS/MST may be differentially expressed in KO#9 (compared to WT and KO#9-REC), this is not reflected in a differential Hippo-dependent phosphorylation of YAP. Finally, it is important to highlight that LATS/MST regulation of YAP leads to phosphorylation-dependent inactivation and retention in the cytosol followed by increased degradation (Zhao et al, Genes Dev 24, 72-85), which is the exact opposite of the effect we observe upon DKK3 expression. In other words, downregulation of LATS/MST1 as observed in KO#9 would theoretically be associated with a release of Hippo-dependent inactivation of YAP and therefore increased YAP activity. However, DKK3-depletion in CAFs is characterised by a strong inactivation of YAP, suggesting that any effect of LATS/MST downregulation as observed in KO.9-CAF is negligible. In our opinion, all these data ruled out any potential participation of Hippo signalling in DKK3-dependent YAP regulation. We discuss now these data more extensively in Page 5, Lines 181-185.

Nevertheless, we have addressed Reviewer#1's concern and now provide additional experimental details. Following Reviewer#1's suggestions, we have performed the proposed experiments to ascertain the role of Hippo signalling in YAP/TAZ activation through DKK3. Noteworthy, we have performed the proposed analyses in WT vs KO.9 CAFs, to be consistent with the rest of the manuscript (i.e. all the previous experiments and comparisons

in Figure 4 & 5 were performed using those cell lines). We observe that almost complete silencing of LATS1&2 expression in CAF1-WT and in KO.9 by RNAi leads in both cases to a significant increase in CAF functions, in agreement with a burst of YAP activity after releasing Hippo breaks. Western blot analyses confirmed that in both cases, knocking down LATS1&2 leads to decreased pS127-YAP levels and a subtle increase in total YAP/TAZ levels. Importantly, the presence or absence of DKK3 does not appear to significantly alter the effect of LATS1&2 silencing in CAF activities or YAP/TAZ levels, excluding a role for Hippo kinases in YAP/TAZ activation through DKK3. These data is presented now in **Figure S4b-d** and discussed in Page 5, Lines 185-190.

- DKK proteins are well known to act as inhibitors of Wnt signaling. Here, the authors show mostly through β -catenin activation that DKK3 acts in favor of Wnt signaling in CAFs. However, β -catenin can be activated by alternative pathways, like PI3K-Akt and MEK-MAPK. Considering that, a similar setting as requested on the above topic (using KO.9 and KO.9 REC cells) should be used to assess the YAP/TAZ levels when treating CAFs with recombinant Wnt and/or the XAV939 inhibitor. In that way, the authors would bring more substantial evidence that DKK3 acts as facilitator of the Wnt pathway in CAFs. The figures supporting DKK3's role in YAP/TAZ activation through the Wnt pathway should be moved to the main text and not left as supplementary.

We thank Reviewer#1 for these suggestions. Regarding the use of XAV939 inhibitor to confirm the functional relevance of Wnt signalling in CAFs, we showed in the original manuscript that XAV939 negatively affected CAF functions, underlying a role for canonical Wnt in regulating CAF activities (**Figure S4f&g**).

To provide additional evidence that DKK3 promotes YAP/TAZ signalling by acting as a facilitator of Wnt signalling in CAFs, we also assessed YAP/TAZ levels and CAF function after Wnt3a stimulation in the presence or absence of DKK3. First, we demonstrate that canonical Wnt3a is able to promote gel contraction in CAFs, whereas non-canonical Wnt5a ligand cannot (**Figure 4g**). Even though DKK3 is not absolutely required for CAFs to respond to Wnt3a, the gel remodelling activities of CAFs after Wnt3a stimulation are diminished in DKK3-null CAFs (i.e. it does not reach levels comparable to wild-type CAFs). This indicates that the presence of DKK3 in CAFs potentiates cellular responses to Wnt3a stimulation. In addition, we now provide data on YAP/TAZ regulation after treatment with Wnt3a ligands in the presence or absence of DKK3. We show that Wnt3a stimulation increases YAP/TAZ activity in CAFs, and that DKK3 is involved in the process (**Figure 4g&h**). Following Reviewer#1's suggestion, these new dataset is presented in the main Figures. These data is discussed in Page 5, Lines 207-214.

Minor comments:

- There are missing supplementary tables; we could not find them?

We noticed this error when trying to retrieve a copy of our manuscript from the Nat Comm server. I am afraid this may be a problem of how Nature handles supplementary data when generating the PDF file for reviewers, as we provided Supplementary Tables as an Excel file. We would gladly provide this again via the Nat Comm editor (or other means as suggested by Nat Comm). In addition, we have tried to upload supplementary tables as PDFs (except for Supp Table 6, which is too big).

-In Figure 2b, typing error – “CAF si#3” should be “CAF1 si#3”.

We apologise for this minor error and thank the reviewer for spotting it. It has now been fixed
-In Figure 3, the letter “o” is referred as letter “p” in the respective legend of the figure.

We apologise for this minor error and thank the reviewer for spotting it. It has now been fixed
- In Figure S4a, typing error – GAPDH is typed as “GAPADH”

We apologise for this minor error and thank the reviewer for spotting it. It has now been fixed

Response to Reviewer #2:

This new MS by Calvo and coworkers represents a very interesting advancement on our understanding of cancer-associated fibroblasts (CAF). Briefly, the authors found that CAF, but not normal fibroblasts, secrete Dickkopf3 (Dkk3), and that stromal Dkk3 expression correlates with tumor malignancy. The authors also found that Dkk3 is required for tumor malignancy in xenotransplant assays, favoring invasive and proliferative abilities of cancer cells. Mechanistically, the authors propose that Dkk3 activates the Wnt pathway by reducing membrane localization of kremen, an inhibitor of the Wnt-coreceptor LRP5/6. In turn, this would allow the activation of beta-catenin, and the non-canonical activation of the Hippo pathway transducers YAP and TAZ. Only YAP/TAZ, but not beta-catenin, are shown to be important for the pro-malignant activities of Dkk3. Although the study is well-conducted and most of the data are quite convincing, there are some issues that need to be clarified/emended before publication:

1) Dkk3 depletion by siRNAs (Fig. 3b) has stronger effects on beta-catenin and on TAZ than the full Dkk3 knockout (Fig. S4a). Please explain this discrepancy, or show better blots for Fig. S4a.

We recognise that perhaps transient knock-down (i.e. RNAi) may appear to have greater effects on β -catenin and TAZ than full knock-out. We reason that a slight variability may be expected from the use of different technical approaches, in particular those involving transient silencing with no adaptation (RNAi) vs stable depletion with potential adaptation (CRISPR knock-out). Importantly, we have extensively validated our results using several CAF models and approaches (siRNA, shRNA, CRISPR/CAS-Recovery, overexpression), as well as both in vivo and with clinical data. All of which underline a positive effect of DKK3 in both β -catenin and YAP/TAZ activities.

2) To convincingly show that the Hippo pathway is not affected by Dkk3 knockout, please reload the blots of phosphor-YAP in Fig. S4a, normalizing by YAP expression; in alternative, please provide quantification of the corresponding bands, normalized by the expression of YAP.

We thank the reviewer for these suggestions. We did not reload the blots normalising to YAP expression as we wanted to illustrate the changes in YAP/TAZ total levels after DKK3 silencing. Instead, we have now quantified pS127-YAP levels normalised to the expression of YAP for this experiment (**Figure S4a**). In addition, we provide quantification of total YAP and TAZ expression (normalised to Gapdh). Overall, there are very marginal variations in the levels of pS127-YAP (normalised to total YAP levels) when DKK3 expression is modulated. This is in striking contrast to the drastic changes in total YAP/TAZ protein expression, which we explored further. We discuss now these in more detail in Page 5, Line 181-184.

Following Reviewer #1 comments (see above), we also provide quantification for LAST1 and MST1 (**Figure S4a**); in addition, we have also performed additional studies to show that Hippo pathway is not altered by DKK3 (**Figure S4b-d**, Page 5 Lines 185-190).

3) The Western blot of Fig. S4c is of poor quality, and should be reloaded. In any case, the reported differences in beta-catenin and TAZ levels are not apparent from these images. Quantifications, normalized to GAPDH, are not adding confidence in this case. It would be more convincing if the authors would be able to present immunofluorescence pictures representative of the data depicted in Fig. S4d, showing that LRP6 knockdown causes nuclear exclusion of YAP.

We apologise for the low quality of the images in Figure S4c. We have now repeated this experiment and provided new blots (**Figure S4f**). Following Reviewer#2's advice, we also show immunofluorescence images representative of YAP localization in control and LRP5&6 knockdown CAFs (**Figure 4c**).

In addition, we now include similar YAP staining images for Kremen1&2 perturbations (**Figure 5k**) and the new data on canonical Wnt activation (**Figure 4h**).

4) The authors must make freely available the microarray gene expression data from normal fibroblasts, normal fibroblasts plus Dkk3, and CAF, either wild-type or Dkk3-KO, that they used in the MS.

We apologize to Reviewer #2 as we inadvertently forgot to update the GEO number on the manuscript before submission. Data related to this study has already been deposited under GSE114056.

REVIEWERS' COMMENTS:

Reviewer #1 (Remarks to the Author):

They have addressed my major concerns.

Reviewer #2 (Remarks to the Author):

The authors addressed all my concerns in full.

Therefore, I support now the publication of this MS.